# Advancements and continued challenges in global modelling and observations of atmospheric ice masses

Patrick Eriksson<sup>1</sup>, Alejandro Baró Pérez<sup>1</sup>, Nils Müller<sup>1</sup>, Hanna Hallborn<sup>1</sup>, Eleanor May<sup>1</sup>, Manfred Brath<sup>2</sup>, Stefan A. Buehler<sup>2</sup>, and Luisa Ickes<sup>1</sup>

**Correspondence:** Patrick Eriksson (patrick.eriksson@chalmers)

**Abstract.** We assess the current status of atmospheric ice mass estimates by comparing global circulation models and global storm-resolving models with satellite observations. The analysis focuses on the frozen water path, which offers a more consistent measure across modelling and observational datasets than cloud ice or other partial quantities. As a reference, we use three retrievals derived from the CloudSat mission. Despite being based on the same input data, these retrievals exhibit a significant spread, and we estimate the uncertainty in overall means to be as high as 40 %. A recently developed machine learning product based on passive observations highly extends spatial and temporal coverage for comparisons, but its accuracy is limited by biases inherited from its training dataset.

The latest generation of global circulation models systematically underestimates frozen water paths compared to the observational benchmark. While the spread among models has narrowed relative to earlier assessments, they still fail to provide consistent representations of regional temporal changes or the annual cycle. Storm-resolving models, which operate at finer grid spacing and resolve convective dynamics explicitly, show a better representation of total ice masses, with a variation among them that's similar to the observational uncertainty. However, several issues were noted, such as apparent deviations from the observations in the spatial structures of tropical deep convection, and that they differ significantly in their relative amounts of cloud ice, snow, and graupel. Together, these findings reveal progress but highlight continuing uncertainties that limit confidence in projections of cloud-related climate feedbacks.

## 1 Introduction

In a broad review, Waliser et al. (2009) found that the status of cloud ice modelling was unsatisfactory, but there were expectations of progress. As a basic illustration of the shortcomings of the time, a number of global circulation models (GCMs) were shown to vary by a factor of 20 in their reported global mean cloud ice mass. On the other hand, there was hope for progress as more advanced treatments of atmospheric ice were being introduced in GCMs, and new satellite sensors were about to provide better observational constraints. Here, we investigate if the expected improvements have been realised by exploring the status of atmospheric ice masses in present sets of global models and satellite datasets.

<sup>&</sup>lt;sup>1</sup>Department of Space, Earth and Environment, Chalmers University of Technology, Gothenburg, Sweden

<sup>&</sup>lt;sup>2</sup>Meteorological Institute, Universität Hamburg, Hamburg, Germany

30

A discussion of atmospheric ice must begin by clarifying terminology. Already the term "cloud ice" is a source of confusion. Inside atmospheric models, cloud ice is normally limited to the fraction of ice particles having a low terminal sedimentation velocity, and can be denoted as "suspended". However, there is no general agreement on a threshold value setting an upper limit in terms of particle fall velocity or size. The remaining ice particles, of larger dimension, are classified as either snow (low density and medium fall speed) or graupel (high density and fall speed). It is noteworthy that the data request document for the latest Coupled Model Intercomparison Project (CMIP6) in effect defines cloud ice as the ice categories considered by the model's radiation scheme. As a consequence, there is still no strict definition of cloud ice found in GCM output.

The vague definition of "cloud ice" was stressed by Waliser et al. (2009) and it was acknowledged that the high deviation between GCMs could partly be attributed to this issue. That is, an equal comparison was not possible and the actual status could not be established. Later studies comparing cloud ice from GCMs with various satellite observation include Eliasson et al. (2011); Li et al. (2012); Jiang et al. (2012); Komurcu et al. (2014); Li et al. (2020).

Compared to cloud ice, a more clear target for a comparison would be to assess the total ice mass. However, this option is of limited applicability for GCMs. This is the case as all, or a large fraction, of the precipitating ice is simply not represented in their output. Accordingly, atmospheric ice masses in GCMs remain difficult to characterise, but the situation is different for the emerging class of global storm resolving models (GSRMs, i.e. models having a horizontal resolution in the order of 5 km). Compared to the high degree of parametrisation applied in GCMs, ice microphysical processes have, to a larger extent, a direct physical representation in GSMRs (Satoh et al., 2008) and the mass of all ice hydrometeor categories are in general prognostic variables.

As also pointed out by Waliser et al. (2009), there are similar considerations when approaching datasets of satellite-based estimates of atmospheric ice masses. One aspect is the sensitivity of the measurement to the particle size distribution (PSD). The rule of thumb is that a remote measurement is most influenced by particles whose sizes are of the same magnitude as the wavelength observed; the measured signal at optical and infrared wavelengths will mainly be governed by the smallest particles, while at microwave wavelengths the larger end of the PSD will dominate the impact (Eliasson et al., 2013).

As an example, the satellite-based cloud radars launched so far, CloudSat and EarthCARE, operate at a wavelength of about 3 mm (94 GHz). In many situations, this wavelength is considerably larger than all the ice hydrometeors in the probed air volume. In this case, the back-scattering is proportional to  $D^6$  (Rayleigh conditions), where D is particle size. The strong dependency on size implies that the measured reflectivity will mainly be generated by the largest particles, despite likely being relatively few; the large particles outshine the small. As a result, additional information is needed to estimate the mass of the non-sensed particle fraction, to map the observed reflectivity to an estimate of the total ice mass.

The same overall logic applies to passive observations: the measurement provides relatively direct information on a fraction of the ice particles. However, the exact size range probed is case-specific, depending on the shape of the local PSD, and no instrument nor retrieval setup is yet capable of isolating e.g. the suspended cloud ice mass. As for models, presently the only clear target for estimates of atmospheric ice contents is the total mass.

Satellite observations are not only limited in terms of particle size, but also with respect to distance into the cloud systems (Eliasson et al., 2011). For active systems (lidars and radars), the emitted signal can be attenuated before reaching the ground

70

level, and for passive systems the optical depth can be too high to effectively probe lower levels. Here, optical and infrared are at an even greater disadvantage, as the attenuation caused by ice crystals increases when decreasing the wavelength, due to the shift towards the impact of smaller (but more numerous) particles discussed above. For the satellite-based cloud radars, strong attenuation is a limitation only for most dense cloud systems. On the other hand, they are affected by surface clutter and there is a blind zone present throughout, extending about 1 km above the surface.

It is also noteworthy that the nomenclature differs between the model and observation communities, such as the meaning of ice water content (IWC) and its column counterpart, ice water path (IWP). These terms are commonly used to name retrieval results and are thus, at least in some satellite datasets, referring to the total ice mass. However, if used in conjunction with atmospheric models, IWC and IWP refer only to the cloud ice fraction. To avoid confusion inside this work, the total of all atmospheric ice is denoted as the frozen water. That is, the frozen water content (FWC) is the sum of the the ice, snow and graupel water contents (shortened as IWC, SWC, and GWC, respectively). Some authors have used the term "total ice" (TIWC) as an alternative to frozen water. The vertical integral of FWC will be denoted as the frozen water path (FWP).

The aim of this work is to assess the present agreement between global models and satellite retrievals. We focus the comparisons on FWP for several reasons. FWP is directly proportional to the latent heat released during the conversion of water vapour to ice. It can be used to estimate precipitation efficiencies, the fraction of hydrometeor condensation that ends up as surface precipitation (Kukulies et al., 2024). The impact on longwave radiation at the top of the atmosphere is, on average, proportional to the logarithm of FWP (Deutloff et al., 2025).

Further, there are now global models (GSRMs) reporting the mass of all ice hydrometeor types and the arguably most reliable satellite retrievals, 2C-ICE and DARDAR, estimate this total mass. In any case, total ice is still the only mass quantity that has a clear definition. DARDAR and 2C-ICE have been used as the reference in many studies, but are generally used separately. Here both these datasets are considered and confronted, to show that they deviate significantly despite using the same measurement input. Besides mean FWPs, the assessment of GSRMs considers the representation of tropical deep convection in terms of organization and diurnal cycles. DARDAR and 2C-ICE do not offer sufficient coverage for addressing these questions and a dataset representing the state of the art for passive retrievals is also applied (denoted as CCIC). As described above, for GCMs having fewer hydrometeor output variables and coarser resolution compared to GSRMs, we are so far limited to considering cloud ice, but we still provide a comparison between the models that participated in CMIP6 to establish a link with earlier similar studies.

#### 2 Data

95

100

#### 2.1 Satellite retrievals

## 2.1.1 Background

Considering how electromagnetic radiation interacts with ice hydrometeors and the spatial resolution achieved by passive and active instruments, estimates of FWP based on cloud radar reflectivities must be considered the most reliable. CloudSat became the first space-based cloud radar when it was launched in 2006 (Stephens et al., 2008). Multiple retrievals exist involving CloudSat, where 2C-ICE (Deng et al., 2010, 2015) and DARDAR (Delanoë et al., 2014; Cazenave et al., 2019) are two widely used ones. Comparisons of 2C-ICE and DARDAR are surprisingly few; exceptions include Deng et al. (2013); Winker et al. (2024); Atlas et al. (2024).

Retrievals based on passive measurements tend to give considerably lower column ice mass values (e.g. Eliasson et al., 2011; Duncan and Eriksson, 2018). This can largely be understood by the fact that 2C-ICE and DARDAR provide estimates of the total ice hydrometeor mass, while the passive datasets tend to only reflect parts of the ice column. Sensors recording optical and infrared radiation are most sensitive to the cloud top layer, with, as mentioned, the signal dominated by small ice crystals. As a consequence, retrieved values have often been interpreted as estimates of the cloud ice mass, roughly corresponding to the suspended fraction of ice particles. Microwave radiation propagates relatively unimpeded through ice clouds and the interaction is dominated by the largest ice particles. This is also the situation for cloud radars, but a higher FWP is required for obtaining a measurable signal with a microwave radiometer. As older passive retrievals have been shown to have a strong bias with respect to CloudSat-based ones, they are excluded from this study.

A new manner to extract FWP from passive observations is to apply machine learning, using a CloudSat dataset as the reference. An early example on such retrievals was SPARE-ICE (Holl et al., 2014), that Duncan and Eriksson (2018) found to agree well with DARDAR in the tropics but to still exhibit a low bias at mid-latitudes. However, SPARE-ICE was only produced for a limited time period. A newer machine learning-based product is CCIC (Chalmers Cloud Ice Climatology, Amell et al. (2024)). By making use of homogenised datasets of geostationary data, quasi-global coverage over decades at high spatial resolution is achieved. As cloud radar data only cover a limited time period, have very low instantaneous spatial coverage and are limited in diurnal sampling (as flown in sun-synchronous orbits), a dataset like CCIC emerges as a potentially very important complement.

## 2.1.2 Included active datasets

Three retrievals involving the CloudSat radar are considered and compared. The two first ones, 2C-ICE and DARDAR, apply the optimal estimation method (OEM, Rodgers (2000)) and also take input from the CALIOP lidar (that flew in tandem with CloudSat). Here we give less attention to the lidar side as it is judged to have a small impact on the overall mean FWP, in comparison to the radar input. As a complement to these established retrieval datasets, radar-only retrievals using an algorithm of "onion peeling" character were also made and are referred to as ARTS onion peeling (AOP). Although both DARDAR

120

135

and 2C-ICE use OEM, the implementations are not the same with the most critical differences being assumptions regarding microphysics.

CloudSat was launched in 2006 into a sun-synchronous orbit and initially provided observations during both day and night, at around 1:45. Due to a battery failure, occurring in 2011, the measurements were limited to the day passage for the later part of the mission until the instrument was turned off in 2023. The horizontal and vertical resolutions of measured reflectivities are about 1.5 and 0.5 km, respectively.

DARDAR version 3.1 (Cazenave et al., 2019) is selected for this study. It is not possible to discern liquid and ice particles from CloudSat reflectivities alone. In the DARDAR algorithm, all reflectivities measured at temperatures below 0°C are assigned to the ice phase. Cloud liquid droplets are too small to generate significant back-scattering at 94 GHz, but the DARDAR approach still assumes that there exist no super-cooled liquid droplets with a size matching drizzle or stronger rain. The ice hydrometeor particle model combines the oblate spheroid representation from Hogan et al. (2012) and the particle size distribution (PSD) from Delanoë et al. (2014) with updated mass-size relation parameters and the calculation of scattering properties assuming totally randomly oriented particles (Cazenave et al., 2019). Only molecular absorption is considered for two-way attenuation of the radar signal.

The 2C-ICE version included is R05. To distinguish ice from liquid in 2C-ICE, cloud phase from the 2B-CLDCLASS-LIDAR (Sassen et al., 2008) product is used. 2B-CLDCLASS-LIDAR locates the presence of liquid water by identifying a sharp increase in the lidar signal followed by a sharp decrease, arising first from lidar's high sensitivity to cloud water and then strong attenuation. If water is present, 2B-CLDCLASS-LIDAR product defines the layer as mixed phase if the cloud layer top is colder than -7°C and the maximum radar backscatter exceeds a temperature-dependent threshold. If 2B-CLDCLASS-LIDAR classifies the region as pure ice, 2C-ICE performs a retrieval. If the layer is classed as mixed phase, 2C-ICE performs retrievals of ice mass from measurements at temperatures below -4°C. As for DARDAR, ice hydrometeors are treated to be totally randomly oriented (Deng et al., 2010), but a mix of particle habits following Baum et al. (2005) is assumed with single scattering properties taken from Hong (2007). Attenuation due to both molecular absorption and ice hydrometeors is accounted for.

As a complement and as a way to test the impact of different assumptions, a module from the Atmospheric Radiative Transfer Simulator (ARTS, Buehler et al. (2025)) was applied. These inversions start at the radar bin with the highest altitude, where an unit transmission can be assumed. The inversion continues downwards, one height bin at a time, using the integrated extinction through the higher bins to estimate the unattenuated reflectivity from the measured one. Retrieving one layer at a time in this fashion has been denoted as "onion peeling" (Rodgers, 2000). We adopt this term and denote the approach as ARTS onion peeling (AOP). In this work, the nominal setup of ARTS assumes that all back-scattering below 0°C is due to ice particles (like DARDAR). Ice hydrometeors are represented as a particle model consisting of a large plate aggregate habit, taken from the ARTS single-scattering database (Eriksson et al., 2018), and the PSD of Field et al. (2007). The attenuation correction considers extinction by molecules and ice hydrometeors. Further details of AOP are found in Appendix A.

It is noteworthy that the PSDs of DARDAR and 2C-ICE both have two free parameters at each altitude, while in the radaronly regime (i.e. below the penetration depth of the lidar) there is only one piece of information per height bin coming from

the measurement. For this reason, a considerable vertical correlation is assumed for one of the two free parameters, to stabilize the retrieval. The AOP takes another approach, using PSDs that have a single free parameter. As a consequence, for two reflectivities, measured at the same temperature, AOP gives the same FWC in both cases, while for 2C-ICE and DARDAR the locally retrieved FWC depends on the full column. This statement neglects attenuation, but its impact is generally low. Common for all three retrievals is that the PSD parameters are temperature dependent.

# 2.1.3 Included passive dataset

The CCIC algorithm has been applied on two "georing" datasets of merged geostationary infrared measurements (Amell et al., 2024). The data used here are based on the NCEP/CPC Merged IR dataset, referred to as CPCIR, covering latitudes between  $\pm 60^{\circ}$ N over 20 years at a 30 min resolution. That is, CCIC offers a continuous view on atmospheric ice masses with close to global coverage over a long time span. The machine learning methodology selected for CCIC, quantile regression neural networks (QRNNs, Pfreundschuh et al. (2018)), ensures that the overall statistical properties of the reference dataset are preserved. Accordingly, statistics, e.g. the zonal mean FWP, of CCIC agrees very well with 2C-ICE (Pfreundschuh et al., 2025) as it is trained on those data. On the other hand, possible biases and limitations in 2C-ICE are inherited and need to be remembered. The retrieval performance at local scales is exemplified in Amell et al. (2024) and further assessment is provided in this work. Both the CPCIR version as well as the GridSat version covering 40 years, show long-term stability; both indicate that the global mean FWP has not experienced any significant change over the last decades. On the other hand, both versions show that changes on regional scales have occurred (Pfreundschuh et al., 2025).

#### 170 **2.2** General circulation models (GCMs)

The Coupled Model Intercomparison Project (CMIP) is an international climate modelling initiative aimed to improve the understanding of past, present, and future climate change using a multi-model framework (Eyring et al., 2016). CMIP has progressed through multiple phases, each introducing enhanced climate model experiment protocols, standards, and data distribution mechanisms. The latest phase with publicly available data output was the sixth phase (i.e. CMIP6). Compared with the previous phase (CMIP5) the CMIP6 models exhibit notable advancements in aspects like spatial resolution, physical parameterization, and the addition of more processes and components of the Earth system (Eyring et al., 2016, 2019; Fan et al., 2020). Furthermore, the number of institutions involved in CMIP6 increased compared to CMIP5, with many contributing several model versions (Eyring et al., 2019).

In this work, we use historical simulations from CMIP6 GCMs to assess their representation of atmospheric ice on global scales. The choice of the CMIP6 GCMs is based on data availability at the time of this study. 16 models are used, some of them with different model versions, resulting in 28 model outputs. Additionally, we complement the analysis with model output from the High Resolution Model Intercomparison Project (HighResMIP phase 1 or HighResMIP1) (Haarsma et al., 2016). That adds ten datasets to the analysis. The horizontal resolution of the simulations spans from 100-500 km for the atmospheric part of the CMIP6 GCMs and 25-100 km for the atmospheric part of the CMIP6 HighRes GCMs (see Table 1).

190

We use the variable *clivi*, defined as the atmosphere mass content of cloud ice (standard name), and ice water path (long name), in this study hereafter to as FWP. This variable includes precipitating frozen hydrometeors only when they affect the calculation of radiative transfer in the model, which is not the case for all models (Table 1).

# 2.3 Global storm-resolving models (GSRMs)

In recent years, with the growth of computational power, the possibility to simulate the atmosphere with a high resolution on global scales has become a reality (Stevens et al., 2019). These non-hydrostatic models typically utilize spatial resolutions on the order of about 5 km and finer. This allows for the explicit simulation of the interaction between small, intermediate and large scales of motion, thereby avoiding the problems caused by the use of parametrizations (e.g., for deep convection) (Randall et al., 2003; Stevens and Bony, 2013; Stevens et al., 2019). The models with the capabilities to perform such simulations on global scales are known as GSRMs.

The Dynamics of the Atmosphere general circulation Modeled On Non-hydrostatic Domains (DYAMOND) intercomparison is the first intercomparison project of GSRMs. Since these models are able to simulate deep convection explicitly, they provide a more robust representation of the climate system and a more direct connection to high-resolution satellite retrievals (Stevens et al., 2019).

So far, two phases of DYAMOND have been carried out: (1st) DYAMOND Summer, and (2nd) DYAMOND Winter. In both of them, the GSRMs were run during 40 days periods: August 1st - September 10th 2016 for the Summer, and January 20th - March 1st 2020 for the Winter phase. The first ten days were the spin-up time for the models. The models did run freely (no nudging towards re-analysis data was applied). Meteorological re-analysis provided by the European Centre for Medium Range Weather Forecasts (ECMWF) was used only to initialize the atmospheric state of the GSRMs. In this study, we use and compare the FWP from the models participating in the DYAMOND Winter phase. Since the GSRMs were freely run, they are not expected to reflect the real weather conditions during the simulated period. Therefore, in our study, the GSRMs outputs are analysed and compared with observations/retrievals from a statistical perspective.

Most DYAMOND models use one-moment microphysics schemes, which classify frozen water into cloud ice, graupel, and snow (Nugent et al., 2022). The data protocol of the DYAMOND intercomparison asked for including the vertical integral of cloud ice, snow and graupel. The models thus provide outputs of vertically integrated cloud ice (IWP), graupel (GWP), and snow (SWP) with output frequencies as can be seen in Table 2. To calculate the FWP we sum up the three vertically integrated hydrometeors mentioned before (IWP, GWP, SWP). That is, the DYAMOND dataset offers a first opportunity to compare the total amount atmospheric ice between different global models, a possibility used by e.g. Nugent et al. (2022); Ćorko et al. (2025). For the second (winter) phase of DYAMOND, 9 of 12 participating models met this suggestion. We chose the 9 DYAMOND models that provide all three parts of FWP (cloud ice, graupel, snow) for our study. In the case of IFS, graupel is not reported separately, but it is included in the snow category (SWP thus reflects both snow and graupel) (Peter Bechthold, ECMWF, personal communication).

|                   | Spatial      |             |
|-------------------|--------------|-------------|
| CMIP6             | resolution   | Falling ice |
| model             | (atmosphere) | included    |
| ACCESS-CM2        | 250 km       | Yes         |
| ACCESS-ESM1-5     | 250 km       | No          |
| AWI-CM-1-1-MR     | 100 km       | No          |
| CESM2-WACCM       | 100 km       | Yes         |
| CMCC-CM2-HR4      | 100 km       | Yes         |
| CMCC-CM2-SR5      | 100 km       | Yes         |
| CMCC-ESM2         | 100 km       | Yes         |
| EC-Earth3-AerChem | 100 km       | Yes         |
| EC-Earth3-CC      | 100 km       | Yes         |
| EC-Earth3-Veg-LR  | 250 km       | Yes         |
| EC-Earth3-Veg     | 100 km       | Yes         |
| EC-Earth3         | 100 km       | Yes         |
| FGOALS-f3-L       | 100 km       | Yes         |
| FIO-ESM-2-0       | 100 km       | Yes         |
| GFDL-ESM4         | 100 km       | No          |
| IPSL-CM5A2-INCA   | 500 km       | No          |
| IPSL-CM6A-LR-INCA | 250 km       | No          |
| IPSL-CM6A-LR      | 250 km       | No          |
| KACE-1-0-G        | 250 km       | Yes         |
| MIROC6            | 250 km       | No          |
| MPI-ESM1-2-HR     | 100 km       | No          |
| MPI-ESM1-2-LR     | 250 km       | No          |
| MRI-ESM2-0        | 100 km       | No          |
| NESM3             | 250 km       | No          |
| NorESM2-LM        | 250 km       | Yes         |
| NorESM2-MM        | 100 km       | Yes         |
| SAM0-UNICON       | 100 km       | Yes         |
| TaiESM1           | 100 km       | No          |
|                   | Cnotial      |             |

|               | Spatial      |             |
|---------------|--------------|-------------|
| CMIP6 HighRes | resolution   | Falling ice |
| model         | (atmosphere) | included    |
| BCC-CSM2-HR   | 50 km        | No          |
| ECMWF-IFS-HR  | 25 km        | Yes         |
| ECMWF-IFS-LR  | 50 km        | Yes         |
| ECMWF-IFS-MR  | 50 km        | Yes         |
| FGOALS-f3-H   | 25 km        | Yes         |
| GFDL-CM4C192  | 100 km       | No          |
| HiRAM-SIT-HR  | 25 km        | No          |
| HiRAM-SIT-LR  | 50 km        | No          |
| INM-CM5-H     | 100 km       | No          |
| MPI-ESM1-2-XR | 50 km        | No          |

**Table 1.** Model specifications of the CMIP6 historical and CMIP6 HighRes hist-1950 simulations used in the analysis. The third column "Falling ice included" reports whether the falling ice is considered in the radiative transfer and included in the FWP. In all cases the configuration variant of the model is r1i1p1f1: realisation 1 (first ensemble member), initialisation method 1 (initial conditions of the piControl experiment), physics 1 (the standard physics setup of the model), forcing 1 (no additional forcing configuration).

|               | Spatial      |           |
|---------------|--------------|-----------|
| Dyamond       | resolution   | Output    |
| model         | (atmosphere) | frequency |
| ARPEGE-NH     |              |           |
| (atmosphere)  | 2.5 km       | 15 min    |
| GEM           |              |           |
| (atmosphere)  | 5.0 km       | 60 min    |
| GEOS          |              |           |
| (atmosphere)  | 3.0 km       | 15 min    |
| GFDL X-SHiELD |              |           |
| (atmosphere)  | 3.0 km       | 15 min    |
| GRIST         |              |           |
| (coupled)     | 5.0 km       | 15 min    |
| gSAM          |              |           |
| (atmosphere)  | 4.0 km       | 15 min    |
| ICON-SAP      |              |           |
| (coupled)     | 5.0 km       | 15 min    |
| IFS-NEMO      |              |           |
| (coupled)     | 4.0 km       | 60 min    |
| MPAS          |              |           |
| (atmosphere)  | 3.75 km      | 15 min    |

**Table 2.** Model specifications of the DYAMOND GSRMs simulations used for the analysis: horizontal resolution (approx. size at the equator) and output frequency. Atmosphere models simulated atmosphere only, coupled models additionally simulated the ocean and land.

## 3 Satellite retrievals

Frozen water path (FWP) from the 2C-ICE, DARDAR, AOP, and CCIC (CPCIR version) datasets (Sec. 2.1) are compared. These four products were selected because they all aim to retrieve the total ice hydrometeor mass, and their results should ideally agree perfectly. This is especially true for 2C-ICE, DARDAR and AOP, as they are all based on measurements by the CloudSat radar, albeit the first two also incorporate the CALIPSO lidar. The overall aim of the comparison is to establish the accuracy of global FWP datasets, to e.g. establish target ranges for the subsequent analysis of various atmospheric models.

The year 2015 was arbitrarily selected for the comparison. During this period, CloudSat measured only during the day part of its orbit and, as a consequence, all data in 2C-ICE, DARDAR and AOP are for a local solar time (LST) of about 13:30. CCIC data have been extracted along the CloudSat orbit by taking data within an interval of  $\pm 15$  minutes around the same LST as CloudSat. Collocations were performed by performing a nearest neighbour interpolation of CCIC onto CloudSat geographical position, where the maximum distance of CCIC from CloudSat is  $0.035^{\circ}$  in latitude and longitude. Only data between  $60^{\circ}$ S

**Figure 1.** Example of collocated retrievals for a scene including deep convection (located between -7.5°N), observed on 28 August 2015. Panels (a) and (b) show the retrieved fields of frozen water content (FWC) for DARDAR and 2C-ICE, respectively. Panel (c) shows retrieved FWP from DARDAR and 2C-ICE, and panel (d) shows retrieved FWP from AOP nominal and CCIC.

and  $60^{\circ}$ N are considered, following the latitude coverage of the CCIC version used. To make the radar-based retrievals more comparable to the horizontal resolution of CCIC and the atmospheric models, the data from 2C-ICE, DARDAR, and AOP were averaged along track over about  $6 \, \mathrm{km}$  (four adjacent radar footprints) before the analysis. This averaging has no impact on regional and zonal means.

#### 3.1 A sample scene

As an introduction to the satellite retrievals, Fig. 1 shows results for an individual scene. The first two panels show retrieved FWC from DARDAR and 2C-ICE to highlight some of their fundamental differences. AOP and CCIC also provide FWC, but are not shown since our focus is FWP.

The FWCs from 2C-ICE and DARDAR agree broadly, but clear differences can be noticed. One example is the widespread thin clouds between 11 and 15 km found in 2C-ICE, but not in DARDAR. The FWP of these clouds reaches  $10\,\mathrm{g\,m^{-2}}$  (third panel). The radar-only AOP retrievals yield zero FWP for these clouds, indicating that the differences between DARDAR and 2C-ICE stem from how the lidar measurements are treated. Given the altitude, these clouds, if present, should consist solely of

260

265

ice. CCIC appears to corroborate the presence of the thin clouds identified by 2C-ICE; however, it should be noted that CCIC is trained on 2C-ICE and may inherit its limitations.

DARDAR and 2C-ICE are more similar regarding the thin clouds found around 7 km. Still, here an opposing example can be noticed. Around  $-2^{\circ}N$  DARDAR reports a FWP exceeding  $100 \,\mathrm{g \, m^{-2}}$ , while 2C-ICE reports only an FWP of  $10 \,\mathrm{g \, m^{-2}}$ .

Further deviations are brought forward by the extensive convective system between -7.5°N and -5°N. In particular, DARDAR reports non-zero FWC down to lower altitudes than 2C-ICE. In the case of DARDAR, all radar back-scattering measured at temperatures below 0°C is assigned to ice hydrometeors, while for 2C-ICE this limit is set to -4°C. On the other hand, the FWC of 2C-ICE tends to be higher in the mid-altitude region of the system, while DARDAR again is higher when reaching its upper parts. Integrated vertically, these differences largely cancel, and the FWPs of 2C-ICE and DARDAR end up being close. AOP gives matching values in the central section of the system but gives lower FWP in the anvil regions on both sides.

A general note on CCIC is its lower horizontal resolution, explaining the relatively smooth variation of FWP. The geostationary data used as input to CCIC have a resolution of about 5 km, but as these observations are highly indirect with respect to the task of determining FWPs, the retrievals incorporate spatial features in the estimation. Technically, this is achieved by making use of convolutions of the input image. As a result, the resolution of the retrieval is deteriorated compared to that of the input data. For the thin clouds discussed above, CCIC tends to match the local peak values of 2C-ICE and thus overestimates the FWP averaged over each cloudy area. For the convective system, CCIC is on the low side compared to other datasets in the core region but gives higher FWPs, e.g., in the anvil around -7.5°N.

# 3.2 Collocation statistics

A more comprehensive view of the agreement between the collocated data is provided by Fig. 2. DARDAR is selected as the common reference for this comparison, but this shall not be taken as an indication that these retrievals are the most accurate.

Already Fig. 1 indicated that there are substantial differences between 2C-ICE and DARDAR, despite being based on the same input data. Fig. 2a shows that the deviation at one location can exceed two orders of magnitude. However, such large deviations are rare, and 68 % of the retrievals are found along the 1:1 line between the two dashed lines. The distribution is somewhat shifted in the direction of higher 2C-ICE values, and the average of these retrievals is higher than that of DARDAR, as explored further below.

In Fig. 2b, it can be seen that there are fewer cases where AOP and DARDAR disagree by orders of magnitude. A contributing factor is that the nominal AOP retrievals apply the same temperature threshold for assigning radar back-scattering to liquid or ice particles as DARDAR (0°C). For the uppermost range of FWPs, the highest densities and the conditional mean are found along the 1:1 line, and there is a symmetric pattern in general. For lower FWP (as reported by DARDAR), there is an asymmetry where higher densities and the conditional mean fall below the 1:1 line, indicating that AOP gives lower means than DARDAR in this range of FWP. This latter observation is partly explained by the fact that AOP does not consider any lidar data and thus misses the contribution from thinner clouds. However, there are situations where AOP is lower by 1 kg m<sup>-2</sup>, and such large deviations show that other factors are also at play.

**Figure 2.** Joint probability distribution functions of collocated FWP retrievals. All panels show DARDAR FWP on the x-axis, with the y-axis showing 2C-ICE in panel (a), AOP nominal in panel (b), and CCIC in panel (c). The data are daytime-only collocations from 2015. Black contour lines represent nine logarithmically spaced levels between  $10^{-5}$  and  $10^3$  m<sup>4</sup> kg<sup>-2</sup>. The red line represents the conditional mean. Black dashed lines represent the 16th and 84th quantile of the conditional distribution of the y-axis retrieval product FWP conditioned on DARDAR FWP.

As expected, there is a high spread between CCIC and DARDAR on local scales (Fig. 2c). This can be attributed to the poorer horizontal resolution of CCIC. However, the CCIC retrievals are successful in the sense that the probability density exhibits a relatively symmetric pattern and that the full range of FWP values is found in CCIC. Except below 3 g m<sup>-2</sup>, the 1:1 line is inside the range spanned by the 16:th and 84:th percentiles. However, CCIC's deviations with respect to DARDAR are not fully symmetric, and the conditional mean lies above/below the 1:1 line below/above about 1 kg m<sup>-2</sup>. The same pattern was found by Amell et al. (2024) when comparing to 2C-ICE, the reference dataset of CCIC.

# 3.3 FWP distributions

Moving to overall statistics, Fig. 3 shows the distribution of FWP inside each dataset. To aid comparison to earlier works, two versions are included. The data in Fig. 3a are normalised to be probability density functions (PDFs), p(FWP), i.e. their integrals are unity:

$$\int_{0}^{\infty} p(\text{FWP}) d\text{FWP} = 1. \tag{1}$$

This type of distribution of FWPs has been reported by e.g. Wu et al. (2009); Eriksson et al. (2014); May et al. (2024). The PDFs in Fig. 3a vary over many orders of magnitude and steadily decrease with FWP. They have the same shape up to about  $5 \text{ kg m}^{-2}$ . Above this, the PDFs of 2C-ICE and AOP match well and are the highest, while the PDF of DARDAR drops off the quickest. CCIC is found in between.

Figure 3. Distributions of retrieved FWP values from daytime observations during 2015 and for latitudes between  $\pm 60^{\circ}$  N. The sampling of the CCIC dataset is centred around CloudSat LTAN with an interval of  $\pm 15$  minutes. All cases (including FWP = 0) are used in the normalisation. Panel (a) shows the overall probability distribution function. Panel (b) shows the logarithmically binned occurrence fractions (using 200 bins between  $10^{-4}$  and  $10^{2}$  kg m<sup>-2</sup>).

In Fig. 3b, logarithmically binned occurrence fractions (log-OFs) are shown,  $o_i(\text{FWP})$ , i.e. their sums are unity:

$$\sum_{i=1}^{n} o(\text{FWP}) = 1. \tag{2}$$

If the bins were equally sized (linearly), these distributions would have the same shape as the PDFs. FWP retrievals have been visualised as log-OFs by e.g. Hong et al. (2016); Sokol and Hartmann (2020); Atlas et al. (2024). When comparing the absolute values of both PDFs and log-OFs it must be considered if all FWP (as done here), or just cases above some threshold, are included. For log-OFs, the absolute values also depend on the selected bin sizes.

The log-OFs vary much less in magnitude. In fact, they are fairly constant between  $1\,\mathrm{g\,m^{-2}}$  and  $1\,\mathrm{kg\,m^{-2}}$ , each varying by less than a factor of two. Compared to the PDFs, differences at lower FWPs stand out more clearly in the log-OFs. For FWP  $

315

320

| Retrieval   | AC(%) | CC(%) | 99.99th |
|-------------|-------|-------|---------|
| DARDAR      | 18.2  | 3.2   | 9.6     |
| 2C-ICE      | 17.7  | 3.7   | 17.6    |
| AOP nominal | 11.4  | 2.4   | 13.1    |
| CCIC        | 21.3  | 3.7   | 13.2    |

**Table 3.** Statistics of the satellite retrievals inside latitudes  $\pm 20^{\circ}$ N. AC is the anvil cirrus fraction  $(0.01 \le \text{FWP} < 1 \text{ kg m}^{-2})$ , CC is the convective core fraction (FWP  $\ge 1 \text{ kg m}^{-2}$ ), and the last column gives the 99.99th percentile in kg m<sup>-2</sup>. Tropical convection has a considerable diurnal variation, in particular over land, and it stressed that these fractions only are derived for the CloudSat passage time of 13:30 (LST).

Thus, the product p(FWP)FWP shows how different FWP-ranges contribute to the mean FWP. It is interesting to note that log-OFs are directly proportional to this product, as the bin widths applied for the log-OFs are proportional to FWP. With this observation, we can deduce from Fig. 3b that FWP cases from below  $1 \, \mathrm{g \, m^{-2}}$  up to several  $\mathrm{kg \, m^{-2}}$  contribute significantly to the mean FWPs. That is, an observation system must cover about four orders of magnitude in FWP in order to provide a basis for correctly estimating global or local mean FWPs.

Sokol and Hartmann (2020) introduced a rough classification of tropical ice clouds: FWPs above  $1 \,\mathrm{kg}\,\mathrm{m}^{-2}$  are treated as convective cores (CC), FWPs below  $10\,\mathrm{g}\,\mathrm{m}^{-2}$  are classified as thin cirrus, and intermediate values as anvil cirrus (AC). Table 3 reports AC and CC fractions according to this scheme, based on the satellite retrievals. 2C-ICE and DARDAR show the most consistent results, but the CC fraction of 2C-ICE is still 16 % higher than DARDAR (in relative terms). It is also noteworthy that the two dataset differ greatly for FWP above  $10\,\mathrm{kg}\,\mathrm{m}^{-2}$ , as indicated by their 99.99th percentile values. The deviations seen in Fig. 3a for high FWP are also valid for the tropical domain.

Including lidar data, as done by DARDAR and 2C-ICE, helps detect thinner clouds, and the low anvil fraction of AOP can be understood as being radar-only. However, the information from the lidar should be of an indirect nature to constrain FWP  $> 1 \text{ kg m}^{-2}$  and the low CC fraction of AOP can not directly be ruled out as being less trustworthy than the ones of 2C-ICE and DARDAR, increasing the possible span for the CC fraction downwards. CCIC has the highest fraction for both AC and CC. This can, at least in part, be attributed to the smearing effect discussed in conjunction with Fig. 1, and CCIC has likely a high bias in both CC and anvil "cloudiness".

#### 3.4 Mean values

The overall means of the datasets are found in Table 4. Since CCIC is trained on 2C-ICE, their mean FWPs are very close, around  $0.14 \,\mathrm{kg} \,\mathrm{m}^{-2}$ . DARDAR and AOP also form a pair, with DARDAR achieving a mean FWP of  $0.116 \,\mathrm{kg} \,\mathrm{m}^{-2}$  and AOP achieving  $0.115 \,\mathrm{kg} \,\mathrm{m}^{-2}$ . The close agreement between DARDAR and AOP is not expected since, despite some similarities in the retrieval setups, AOP applies a particle model that is entirely independent of the one used in DARDAR.

The zonal means found in Fig. 4 show that the agreement in mean FWP between 2C-ICE and CCIC, and between the DARDAR and AOP, are not limited to the overall mean. This pattern is also seen at each latitude range. The same applies to

| Dataset     | Mean FWP [ $kg m^{-2}$ ] |
|-------------|--------------------------|
| DARDAR      | 0.116                    |
| 2C-ICE      | 0.144                    |
| AOP nominal | 0.115                    |
| CCIC        | 0.143                    |

**Table 4.** Overall means of FWP between  $60^{\circ}$ S and  $+60^{\circ}$ N.

Figure 4. Zonal mean of FWP retrieved from daytime observations during 2015. The sampling of CCIC is centred around CloudSat LTAN with intervals of  $\pm 15$  minutes.

regional means. Figure 7 of Pfreundschuh et al. (2025) shows that the spatial patterns of mean FWP from 2C-ICE, DARDAR and CCIC are very similar, and they only differ in the overall mean value. AOP also fits into this picture (not shown).

## 330 3.5 Sensitivity tests

The AOP framework is used here to test the sensitivity to several assumptions employed in the radar retrieval schemes. Performed tests and obtained changes in global mean FWP are reported in Table 5. For background information on AOP, see Sec. 2.1.2 and Appendix A.

The first tests refer to changes in the particle model. Changing from the nominal large plate aggregate (LPA) to two other aggregate particle habits from Eriksson et al. (2018) affects the mean FWP drastically. The large column aggregate (LCA) gives less backscattering for a given particle mass. Reversely, a measured reflectivity then maps to a higher ice mass content and the global mean FWP more than doubles compared to when using LPA. Switching to the large block aggregate (LBA) has the opposite effect, as this habit constitutes a harder radar target, resulting in an FWP decrease of 38 %.

The second aspect of the particle models in AOP is the assumed PSD. Replacing Field et al. (F07, 2007) with Delanoë et al. (D14, 2014, used as in May et al. (2024)) was found here to have a small effect, but this is not a general result since there could be a significant effect when used in conjunction with other habits. This is illustrated by Fig. 6 of Ekelund et al. (2020). This

| Habit | PSD | LWC | T limit | aARO | ΔFWP   |
|-------|-----|-----|---------|------|--------|
| LPA   | F07 | No  | 0°C     | 0 dB | -      |
| LCA   | F07 | No  | 0°C     | 0 dB | +110 % |
| LBA   | F07 | No  | 0°C     | 0 dB | -38 %  |
| LPA   | D14 | No  | 0°C     | 0 dB | +2 %   |
| LPA   | F07 | Yes | 0°C     | 0 dB | +12 %  |
| LPA   | F07 | No  | 0°C     | 1 dB | -23 %  |
| LPA   | F07 | No  | -5°C    | 0 dB | -28 %  |

**Table 5.** Change in overall mean FWP,  $\Delta$ FWP, as a result of changes to the AOP retrieval scheme. The change in FWP is given as a percentage relative to the AOP nominal setup, i.e. the setup described in the first row of the table. The columns list in order: the particle habit applied, the particle size distribution (PSD) applied, if liquid water content (LWC) is considered in calculation of pulse attenuation, ice/liquid temperature limit (T), size of approximation for azimuthally random orientation (aARO) and the resulting  $\Delta$ FWP. Remaining acronyms are described in the text.

figure displays AOP retrievals for combinations of three PSDs and eight habits, serving as a complement to the results derived here.

In reality, both habit and PSD vary from position to position. However, satellite observations provide limited constraints on this variability, and the selection of particle model essentially boils down to finding the best average one. The nominal AOP particle model is primarily motivated by its use in RTTOV-SCATT to represent snow (Geer, 2021), making it widely used for assimilating passive observations at frequencies similar to those used by satellite-based cloud radars (94 GHz). The combination of F07 and LPA was singled out based on global observations, but the process assumed that the atmospheric data used (from ECMWF) had no biases, making the optimisation model specific. A similar study by Fox (2020), using another atmospheric model and restricted geographically, instead found F07 and LCA as the best particle model. The same combination was also found to be the best by Ekelund et al. (2020), based solely on satellite observations but limited to tropical conditions. The latter two studies found that LBA, in combination with F07, was a poorer option than the nominal particle model. For example, it deteriorates the agreement with the reflectivity-FWC relationship derived by Protat et al. (2016), compared to using LPA and LCA (Ekelund et al., 2020).

In summary, the question of a "one size fits all" ice particle model for the retrievals of concern remains an open question; however, the judgement is that the particle model selected for AOP (LPA) is more likely to yield a low bias than a high one.

Liquid cloud droplets are too small to cause significant reflectivity (for CloudSat), but they still attenuate the radar pulse; however, this effect is neglected by DARDAR, 2C-ICE, and the nominal AOP setup. This results in a tendency towards under-

estimation of ice masses. Including LWC from ERA5 (Hersbach et al., 2020) in the two-way attenuation of the radar pulse increases the overall mean with 12 %. The found value depends on how well ERA5 represents supercooled water, a quantity known to be poorly represented in atmospheric models (Komurcu et al., 2014; Korolev et al., 2017). There is also uncertainty in the microwave attenuation of supercooled water (Lonitz and Geer, 2019). The impact of neglecting LWC is the strongest at low latitudes.

All three retrievals involve non-spherical ice particles in their calculation of radar backscatter, but they all consider these to be totally randomly oriented (TRO). This is a simplification: ice hydrometeors tend to align their largest dimension horizontally (e.g. Hogan et al., 2002; Matrosov et al., 2005; Gong and Wu, 2017; Brath et al., 2020). Azimuthal random orientation (ARO) should still apply. By assuming TRO instead of ARO, the backscattering for a given ice mass is under-estimated, causing a high bias in the retrievals. Our test with AOP resulted in a 23 % decrease in FWP, assuming that particle orientation matches a change of 1 dBZ. Hogan et al. (2006) found that  $\log_{10}(\text{FWC})$  is linearly proportional to 0.06dBZ. This relationship gives a lower sensitive, -15 % dB<sup>-1</sup>. Marchand et al. (2013) found a median orientation effect of 2.4 dB, but this was based on data from a single measurement campaign.

As mentioned, cloud droplets do not generate a significant backscattering, but precipitation-sizes drops are still present at temperatures below 0°C in updrafts. On the other hand, ice is found above 0°C in falling melting particles. This raises a question of definition: shall ice in melting particles be included in FWP or not? Since including this ice fraction would further complicate the overall assessment, we set it aside for now. As for the particle model, the local variation can not be resolved, and the retrievals apply a global mean temperature threshold for delineating reflectivities generated between ice and (larger) liquid particles. The last test reported in Table 5 refers to changing the threshold temperature from 0°C to -5°C, a value that may be more realistic considering the discussion above and closer to the assumption in 2C-ICE (-4°C). This change decreases FWP by 28 %.

Adding LWC increases FWP, but considering particle orientation and switching to a more realistic temperature threshold (ignoring melting ice) yields larger, opposing effects. Accordingly, the combined treatment of these three aspects in the nominal version of AOP likely results in a high bias. The bias is presumably significant.

## 3.6 Retrieval trueness

To simplify the following discussion, DARDAR is selected as a reference. By Table 4, we find that the 2C-ICE mean FWP is about 24 % above that reference level. Putting some weight on the fact that DARDAR and AOP agree on mean FWP, but not ruling out 2C-ICE, we centre our uncertainty range at +10 % (with respect to DARADR means).

It is acknowledged that studies comparing DARDAR and 2C-ICE retrievals with airborne measurements do not observe any obvious biases (Deng et al., 2013; Heymsfield et al., 2017). This is encouraging, but these assessments are done for individual levels, typically above the mixed-phase region, and considerable possible biases in FWP can not be ruled out. The tests reported in Sec. 3.5 indicate a possibility of much higher FWP due to the selection of particle model, while the treatment of particle orientation and temperature threshold leaves room for considerably lower FWP.

The overall conclusion is that it is impossible to deduce strict bounds on the systematic errors. It can only be concluded that they are possibly substantial. For simplicity, we follow Austin et al. (2009) and assume an uncertainty range around 40 %. That is, we will below indicate the possible range of mean FWP as being 0.7 to 1.5 times the DARDAR mean. We cannot assign a likelihood to our uncertainty estimate, but we notice, despite not covering all the changes reported in Sec 3.5, that the ratio between the upper and lower bounds is approximately a factor of two. Stein et al. (2011) found a similar ratio.

#### 3.7 Outlook

420

A second cloud radar has now been launched, EarthCARE (Wehr et al., 2023). It is a 94-GHz single-frequency radar, like CloudSat, and retrievals remain highly sensitive to the assumed particle model. However, there are several improvements. This new radar has a higher sensitivity, extending the retrieval coverage to lower FWPs, and it has the capability to measure vertical velocities, providing additional input for retrieval schemes.

CCIC demonstrates that machine learning unlocks new possibilities, and we anticipate seeing improved FWP estimates based on existing passive measurements. For example, continuous production using the SPARE-ICE (Synergistic Passive Atmospheric Retrieval Experiment-ICE) algorithm (Holl et al., 2014) has just started. SPARE-ICE is a dataset of FWP retrievals derived exclusively from passive microwave and infrared observations using the Microwave Humidity Sounder (MHS) and the Advanced Very High Resolution Radiometer (AVHRR) on board of NOAA-18, NOAA-19 and the MetOp satellites. The retrievals are performed using machine learning, with the model trained on 2C-ICE data collocated with MHS and AVHRR observations from NOAA-18.

Passive retrievals will also benefit from data in a new wavelength ( $\lambda$ ) region – the sub-millimetre. The Ice Cloud Imager (ICI) is an upcoming mission in this direction, dedicated to measuring ice hydrometeor properties (Eriksson et al., 2020). This sensor, scheduled for launch in 2026, will conduct observations at higher microwave frequencies than those currently used, thereby enhancing sensitivity to ice cloud masses (Evans and Stephens, 1995; Jimenez et al., 2007). ICI will have channels up 664 GHz ( $\lambda = 0.45$  mm), but the Arctic Weather Satellite (AWS, Eriksson et al. (2025)), launched in 2024, already provides a first glimpse of ice clouds at the lower end of the sub-millimetre range with four channels around 325 GHz ( $\lambda = 0.9$  mm).

Retrieval products based on EarthCARE and AWS are still in development. Some early results are exemplified in Fig. 5. The EarthCARE results are taken from the radar-only CPR\_CLD\_2A product (Mroz et al., 2023), baseline BA. These retrievals ignore reflectivities below -21 dBZ as the Doppler information there is less reliable. In addition, no retrievals are so far made if the column is classified as having characteristics of convection. For simplicity, the FWP for missing data was set to zero, explaining the comparably low mean values in the Tropics.

The AWS retrievals were performed at Chalmers University of Technology, following May et al. (2024). The AWS zonal mean remains within the grey area throughout. It is noteworthy that the AWS algorithm does not involve any other retrievals (in contrast to CCIC); instead, the machine learning is based on physical simulations. The preliminary data from EarthCARE and AWS are included here only as an outlook; changes in both retrieval systems are expected.

As both CCIC and SPARE-ICE are trained on 2C-ICE data it is not surprising that their zonal means align closely south of 15° N. However, there is a clear disagreement for the remaining part of the northern hemisphere, showing that machine

**Figure 5.** Zonal mean of FWP retrieved from currently operational sensors: EarthCARE and AWS (July and August, 2025) and CCIC (July, 2025). The CCIC data have been sub-sampled to the local solar times observed by AWS (approx. 10:30 and 22:30). The grey area represents the expected range according to older satellite observations (Sec. 3.6), based the zonal mean of DARDAR for July and August during 2007 to 2010.

learning based on collocations has specific issues to consider. For this northern range, there is instead high agreement between SPARE-ICE and AWS.

#### 4 General circulation models

In the following, we compare the FWP for a selection of CMIP6 and CMIP6-HighRes models (historical simulations). The variable used for the analysis (*clivi*) refers to the atmosphere mass content of cloud ice and does not always include precipitating frozen hydrometeors (Sec. 2.2), i.e., snow and graupel. In other words, not all ice is represented in the FWP for all models, and we can expect lower values from the CMIP6 analysis compared to satellite retrievals. However, we still compare the observations following Waliser et al. (2009), Jiang et al. (2012), Li et al. (2020), and others. More details on the specific models and the inclusion of precipitating hydrometeors can be found in Table 1. Note that in the following, the overall assessment is based on global means and not limited to 60° S - 60° N (as done for the satellite retrievals).

### 4.1 Mean values

We first explore global mean values. Figure 6 shows that, in both groups of models, the mean FWP is within similar ranges: from 0.012 to  $0.082\,\mathrm{kg\cdot m^{-2}}$  in the CMIP6 historical simulations, and from 0.013 to  $0.070\,\mathrm{kg\cdot m^{-2}}$  in the CMIP6-HighRes simulations. This means that the models with the largest FWPs have values around 6 to 8 times higher than those with the smallest FWPs. Waliser et al. (2009), in a similar comparison using CMIP3 models, found differences of up to a factor of 20 between the models with the highest and lowest FWP values (Fig. 1d in Waliser et al. (2009)). However, this large spread was primarily due to two models with markedly high FWPs. When those outliers were excluded, the remaining models showed

**Figure 6.** Global means of monthly FWP means during the period 2000-2014 for the CMIP6 historical simulations (bars in blue with stripes descending), and CMIP6 HighRes hist-1950 simulations (bars in orange with stripes ascending). Models including falling ice in their FWP are marked with an asterisk (\*).

differences of about a factor of 6 – comparable to our findings. Moreover, although not explicitly quantified in Waliser et al. (2009), the global mean FWPs of these models appear to fall within a similar range to those we found in our study. It is noteworthy that there is no clear separation between models that just include cloud ice in the reported FWP and those that also add falling ice. For example, the two model categories are represented at both ends of the FWP range (Fig. 6).

To further explore the differences, we calculate the FWP zonal means and compare these with satellite retrievals (Fig. 7). Most models participating in the CMIP6 and CMIP6-HighRes underestimate the FWP when compared to satellite retrievals. In the historical simulations (Fig. 7a), there is a large spread in FWP values across the mid and high latitudes, with some of the models approaching or falling within the satellite retrieval range (e.g. IPSL-CM5A2-INCA, ACCESS-ESM1-5, ACCESS-CM2 and KACE-1-0-G). On the other hand, the values are clustered below 0.05 kg·m<sup>-2</sup> in the tropics, and are substantially smaller than the satellite retrieval. The only exception is the FGOALS-f3-L model, whose values in the tropics and subtropics fall within the satellite range. In the CMIP6-HighRes simulations (Fig. 7b), the high-resolution version of the model FGOAL (FGOALS-f3-H) also produces FWPs values in the tropics and subtropics that fall within the satellite range. Both FGOALS simulations (for CMIP6 and CMIP6-HighRes) produce very similar FWPs. The remaining models appear to be divided into two distinct groups, with one group —HiRAM-SIT-HR, HiRAM-SIT-LR, and GFDL-CM4C192—exhibiting higher mean FWPs across all latitudes and aligning more closely with satellite retrievals than the other models. Another interesting feature is that the FWPs of the simulations performed with the same model but at different resolutions (e.g., ECMWF and HiRAM) do not differ substantially. Furthermore, the combined information from Table 1 and Fig. 7b does not suggest an improvement in the simulated FWPs when the resolution increases from 100 to 25 km. This is not particularly surprising since at those resolutions

**Figure 7.** Zonal means of monthly FWP means during the period 2000-2014 for (a) CMIP6 historical simulations, and (b) CMIP6 HighRes hist-1950 simulations. The grey area represents the expected range according to satellite observations (Sec. 3.6), based the zonal mean of DARDAR for the years 2007-2010.

many processes in these models (e.g., deep convection and ice-phase processes) still need to be parametrized and the model physics is the same for the models participating in both, CMIP6 and CMIP6-HighRes.

Previous research comparing GCMs with observations has identified several of the features we highlighted earlier in our analysis. For instance, Komurcu et al. (2014) also reported an underestimation of FWP in GCMs (from the IPCC AR5) compared to observations, which is consistent with the FWP range they obtained (as described above). Moreover, Eliasson et al. (2011) found that, over the tropical oceans, the FWP in half of the six GCMs from the IPCC AR4 they evaluated fell below or well below the uncertainty range of satellite retrievals. In our case, this underestimation applies to most models across the entire tropical region, including both land and ocean areas.

Despite the large variability of FWP in between the models, we can still conclude that none of the models overestimate FWP compared to the satellite retrieval, and therefore none of the models can be falsified. However, it must be considered unlikely that non-reported ice masses can explain the gap to the satellite range for all models.

**Figure 8.** Trends in FWP from 1980 to 2014 for (a) FGOALS-f3-L (CMIP6 historical), and (b) FGOALS-f3-H (CMIP6-HighRes hist-1950). The trends based on the monthly mean values in each model grid cell for the full period. Red-shaded regions correspond to increases in FWP, whereas blue-shaded regions correspond to decreases in FWP.

#### 4.2 Trends

Time series of monthly global mean FWP covering 35 years are found in Fig. B1. Linear fits of these time series give trends between -1.0 and +0.5 %/decade. This lack of clear trends is in agreement with the CCIC data record, also indicating a basically constant global mean FWP since 1983 (Pfreundschuh et al., 2025). On the other hand, CCIC and other datasets show that there have been clear regional trends in mean FWP. For the period 2003 - 2023, Pfreundschuh et al. (2025) found a surprisingly high agreement for such trends between ERA5 (Hersbach et al., 2020), CCIC and two other satellite retrievals (MODIS and PATMOS-x), with substantial areas having trends in excess of  $\pm 20$  %/decade. The same analysis for 1983 - 2023 gave a much less consistent view (due to inhomogeneities in the input observations). However, the overall picture remains that regional trends have also occurred over this more extended time period.

Here, we disregard the exact value of regional trends and instead investigate the agreement between GCMs with respect to such trends. To minimise the impact of differences in global mean FWP, trends are reported relative to local means. This approach was adopted by Pfreundschuh et al. (2025), and as mentioned, a high level of agreement was obtained for trends between CCIC and ERA5, even though the latter dataset does not report all ice masses.

As an introduction, the trends for the standard and high-resolution versions of the FGOALS model are shown in Fig. 8. This choice is based on the fact that, in terms of the FWP zonal mean, this model shows the best agreement with the satellite retrievals (Fig. 7). Similar spatial patterns are observed in the decadal FWP trends of FGOALS-f3-L and FGOALS-f3-H. However, FGOALS-f3-H exhibits more pronounced extreme values—both positive and negative—than FGOALS-f3-L in certain regions, such as the northeastern tropical Atlantic (negative trends) and the Arabian Sea (positive trends).

Regional FWP trends for all models are found in Figs. B2 and B3. There is a clear spread in the trends between the models. However, two main groups emerge. Most model configurations yield trends with magnitudes and spatial structures similar to

Figure 9. Annual cycles of relative FWPs (FWP<sub>relative</sub>), calculated for each model as the ratio of monthly mean FWP to the mean FWP over 1980–2014. Results are shown for (a) CMIP6 historical simulations and (b) CMIP6 HighRes hist-1950 simulations. The annual cycle of FWP<sub>relative</sub> for CCIC over 1998–2023 is also shown in both panels for comparison.

those of FGOALS discussed above. This group can be claimed to be in general agreement with the indications on 40-year trends derived by Pfreundschuh et al. (2025) (neglecting noticeable artefacts in the observations). This is in contrast to a second group of model results characterised by having widespread positive trends, counterbalanced by more local negative trends in tropical areas having high FWP. Some models end up between these main patterns. There is no clear link here to model characteristics. Both standard and high-resolution GCMs are represented in both groups. The same for models including falling ice or not. The same model can end up in both groups depending on the configuration (e.g. EC-Earth3).

## 4.3 Annual cycles

Variations of global mean FWP with a period of one year can be discerned in Fig. B1. To investigate this further, long-term mean annual cycles were derived and are displayed in Fig. 9. CCIC indicates a semi-annual variation with maxima and minima at about 2 % above and below the annual average. The maxima occur about one month before the summer/winter solstices. The annual cycles of the GCMs vary. Compared to CCIC, the amplitude of the models' cycles is, in general, higher, with peak-to-peak amplitudes of up to about 15 %, but there are also some models with weaker cycles. Most models exhibit a semi-annual pattern similar to CCIC, but there are also models where the peak during boreal spring dominates. For the boreal spring, there is also a tendency for the models to peak about one month earlier (April-May) than CCIC (May-June). In summary, most models provide a fair representation of the annual cycle in global mean FWP, but some notable deviations are evident.

| GSRM   | Mean FWP [ $kg m^{-2}$ ] |
|--------|--------------------------|
| ARPEGE | 0.100                    |
| GEM    | 0.072                    |
| GEOS   | 0.113                    |
| GFDL   | 0.087                    |
| GRIST  | 0.079                    |
| GSAM   | 0.050                    |
| IFS    | 0.136                    |
| MPAS   | 0.101                    |
| ICON   | 0.075                    |

**Table 6.** Means FWP between 60°S and 60°N of considered GSRM model runs (covering Feb 2020).

### 5 Global storm-resolving models

In this section, FWP in global storm-resolving models (GSRMs) is assessed. The aim is to explore the general capabilities of this new type of models. Details of individual models are brought forward to indicate issues to be resolved, not to rank the models in any manner. The GSRM data is taken from the DYAMOND model intercomparison project and covers February 2020 (Sec. 4). Following the satellite section, we limit the overall assessment to  $60^{\circ}$  S -  $60^{\circ}$  N. This is complemented with a dedicated section on the Tropics ( $20^{\circ}$  S -  $20^{\circ}$  N).

## 5.1 Overall statistics

The (quasi-global) mean FWPs of the GSRM models are shown in Table 6. Given that these models incorporate advanced features intended to improve the representation of the climate system compared to the GCMs (see Section 2.3), we expect their FWPs to show better agreement with observations. Only IFS is inside the range covered by the satellite retrieval means in Table 4, all other models have lower mean FWP. Considering that Table 4 reports annual means while the mean of February (for CCIC) is at about 3 % lower (Fig. 9), also GEOS can be considered to have a good agreement with the satellite retrievals. When also applying the broader target range motivated in Section 3.6, which sets a lower limit of approximately 0.079 kg m<sup>-2</sup>, four additional models fall within this range: ARPEGE, GFDL, GRIST, and MPAS. Among the remaining models, GEM and ICON lie relatively close to this lower bound, while GSAM reports the lowest value at 0.050 kg m<sup>-2</sup>.

For a comprehensive view of the spatial distribution, Fig. 10 compares the model ensemble mean FWPs with corresponding CCIC satellite retrievals. Global distributions for the DYAMOND summer phase (winter phase used in this work) are found in Fig. 1 of Ćorko et al. (2025).

As discussed above, the model ensemble exhibits a lower global mean FWP compared to CCIC. Nevertheless, disregarding this offset in the mean, the agreement shown in Fig. 10 can be considered strong. The spatial patterns of mean FWP in the models generally align well with those derived from observations, despite the fact that the GSRMs were not nudged to

**Figure 10.** Gridded (1° resolution) mean FWP for February 2020, for CCIC (top panel) and averaged over the nine considered GSRMs (lower panel).

reproduce the actual weather during the analysed period. For instance, the signatures of the Intertropical Convergence Zone (ITCZ), as well as the Andean, Zagros, and Himalayan mountain ranges, are strikingly similar in both panels. On the other hand, the model ensemble shows more extensive regions of very low mean FWP across the tropics and subtropics.

Zonal means from the GSRMs are shown in Fig. 11. The majority of models produce values within the satellite-derived range but tend to cluster near its lower bound, or even fall below it, consistent with the earlier discussion on global means. The GSAM model consistently exhibits the lowest mean FWP across most of the latitudes. Interestingly, Atlas et al. (2024) found that, in GSAM, the SAM1MOM microphysics scheme—also used in the DYAMOND simulations—produced the lowest FWP among several available microphysics schemes. This suggests that the selection of this microphysics scheme is likely the primary factor contributing to the relatively low FWP values in GSAM's DYAMOND simulations. In the tropics, IFS stands out by having a mean FWP about double as high as any other model around 10° S. IFS shows a similar behaviour in the DYAMOND summer phase (Ćorko et al., 2025, Fig. 2). Compared to the GCMs, the spread among the GSRMs is relatively small. This is partly due to the consistent definition of FWP applied across all models, but it also represents a positive indication of their ability to bracket the mean FWP of Earth's atmosphere. Notably, if the highest and lowest values in each latitude bin are excluded, the remaining model spread closely matches that of the satellite retrievals.

Figure 12 provides an overview of individual FWP values. Aside from discretization artefacts, the models exhibit similar distributions, generally peaking around  $20 \text{ g m}^{-2}$ . The main exception is GFDL, which displays a double-peak structure, with

**Figure 11.** Zonal mean FWP of the GSRM models (lines). The grey area represents the expected range according to satellite observations (Sec. 3.6), based the zonal mean of DARDAR for January to March in the years 2007-2010. The primary peak of the IFS-model is cut off in an attempt to improve visual clarity for all other models. It has a maximum FWP value of  $0.46 \, \mathrm{kg \, m^{-2}}$  at  $9.5^{\circ}$  S.

maxima around 0.1 and 5 g m<sup>-2</sup>. The same bin size has been applied in Figs. 12 and 3b and the results are comparable. Model and observational distributions agree most closely at high occurrence fractions, where they exhibit similar peak values, while most models show higher fractions than the observations around FWP  $\approx 10^{-4} \text{ kg m}^{-2}$ .

As explained in Sec. 2.3, the output of the GSRMs distinguishes between ice water path (IWP), snow water path (SWP) and graupel water path (GWP), with the FWP defined as the sum of these three components. This detailed output allows to compare the partitioning among ice-phase hydrometeors in the GSRMs, an analysis that is not possible with the GCMs due to the absence of these specific output variables. The IWP, SWP and GWP fractions for quasi-global data are visualised in Fig. 13. These fractions were also calculated over other latitude ranges, but since only small variations were found these results are not presented. The most clear difference between fractions between the extra-tropics and the tropics, is a somewhat higher graupel fraction in the later case, at the expense of cloud ice. An exception is the GSAM model, that exhibited the opposite behaviour. The snow fractions stayed unaffected by the latitude limits applied (with the exception of IFS that does not distinguish between snow and graupel).

Although the fractions of FWP experience only minor variations across latitude changes within each model, there is a clear lack of agreement between the models. The fractions vary as follows: snow 17-95 %, graupel 3-32 %, and ice 1-64 %. There is no obvious relationship between the category fractions and mean FWP. For example, GSAM and GEOS have similar fractions, but GSAM is the model with lowest FWP and GEOS is around or above average in zonal mean values.

Figure 12. Logarithmically binned occurrence fractions (log-OF) of each model's FWP between  $60^{\circ}$  S and  $60^{\circ}$  N (using 200 bins between  $10^{-4}$  and  $10^{2}$  kg m<sup>-2</sup>). The spikes for some models are caused by discretization of the output.

Figure 13. Relative fraction of ice, snow and graupel to the total frozen water, for each GSRM model. Compiled using all data between  $60^{\circ}$  S and  $60^{\circ}$  N. For IFS, snow incorporates the graupel category.

In summary, Fig. 13 illustrates that there is no common view among the GSRMs on the fractioning between cloud ice, snow and graupel. It is outside the scope of this work to explore this further. However, presumably, both basic definitions of the meaning of these ice categories and direct model shortcomings contribute to this poor agreement.

| GSRM   | AC(%) | CC(%) | 99.99th |
|--------|-------|-------|---------|
| ARPEGE | 23.4  | 2.6   | 16.3    |
| GEM    | 16.5  | 1.4   | 14.5    |
| GEOS   | 14.2  | 2.0   | 52.2    |
| GFDL   | 21.2  | 2.0   | 25.9    |
| GRIST  | 13.9  | 1.9   | 17.7    |
| GSAM   | 16.2  | 1.1   | 14.7    |
| IFS    | 16.6  | 3.1   | 35.3    |
| MPAS   | 28.4  | 2.1   | 14.5    |
| ICON   | 20.0  | 1.7   | 12.0    |

**Table 7.** Statistics of GSRM FWP inside latitudes  $\pm 20^{\circ}$  N. AC is the anvil cirrus fraction  $(0.01 \le \text{FWP} < 1 \text{ kg m}^{-2})$ , CC is the convective core fraction (FWP  $\ge 1 \text{ kg m}^{-2}$ ), and the last column gives the 99.99th percentile in kg m<sup>-2</sup>.

# 5.2 The tropical region

The higher horizontal resolution of GSMRs, in comparison to GCMs, is expected to be especially beneficial for the representation of tropical deep convection, and, accordingly, many studies making use of the DYAMOND database are focused on the tropical region (e.g. Christensen and Driver, 2021; Su et al., 2022; Nugent et al., 2022; Turbeville et al., 2022). In this section we follow along the same line and present statistics related to the impact of deep convection.

The high end of the distribution of tropical FWP of each model is summarised Table 7, following the same analysis of the satellite retrievals in Sec. 3.3. There is a considerable spread among the models in their 99.99th percentile, with values from  $12.0 \text{ kg m}^{-2}$  (ICON) to  $52.2 \text{ kg m}^{-2}$  (GEOS). The corresponding range for the satellite retrievals is  $9.6 - 17.6 \text{ kg m}^{-2}$ , but a low bias in these values is likely due to the high attenuation of the radar signal at very high FWPs. Accordingly, all the models 99.99th percentile can be considered as reasonable. In any case, none of the models exhibits an apparent deficiency in simulating very high FWP values.

Despite reaching FWP in excess of  $10 \text{ kg m}^{-2}$ , the models tend to have low CC fractions. None surpasses the CC fraction derived from DARDAR of 3.2 %, and they fall even further below the 3.7 % reported by both 2C-ICE and CCIC (Table 3). The closest is IFS at 3.1 %. The CC fraction of AOP is considerably lower (2.4 %) but, beside IFS, only ARPEGE is above this value. The CC fractions correlate to some extent with the models' overall mean FWP. The models AC fractions vary with a factor of 2, between 14 and 28 %, but they are fairly well centred around the observations.

The remainder of this sub-section is focused on the organization of convective areas. CC has been previously defined as  $FWP \ge 1 \text{ kg m}^{-2}$ , as introduced by Sokol and Hartmann (2020) and subsequently adopted by e.g. Nugent et al. (2022); Turbeville et al. (2022); Bolot et al. (2023). However, as discussed above, applying this limit results in varying CC area fractions between the models, with an overall low bias with respect to the observations. As a consequence, below we instead set the

**Figure 14.** Distribution of number of convective core objects, inside 20° S to 20° N for each GSRM model and CCIC (Feb 2020). The grey areas are the probability distributions of object numbers, between model and retrieval time steps. A corresponding box-plot (representing the 0th and 100th percentile as whiskers, the 25th, 50 and 75th percentile as a box and the median as a red line) is shown underneath. Objects are defined as the set of adjacent grid cells (including diagonals) whose FWP value is in the top 3 % for the dataset of concern.

Figure 15. The probability distribution of convective core object areas. Considered area and definition of objects as in Fig. 14.

threshold at 97th percentile of the FWP distribution of each model (for tropical 24 hour data). The 97th percentile is selected as the satellite observations indicate a CC fraction of 2.4 to 3.7 % (Table 3). This normalised threshold is denoted as  $CC_n$ .

Figure 16. The diurnal variation of data exceeding the  $CC_n$  threshold, of the GSRM models and CCIC. The diurnal average of all datasets is 3 %, as stipulated by the definition of  $CC_n$ .

Figures 14 and 15 provide statistics of convective objects defined according to CC<sub>n</sub>. The strength of CCIC here emerges, since it is the only satellite derived dataset providing the spatial coverage needed for this analysis. The only model that exhibits statistics comparable to CCIC is IFS, which closely matches the observations in both the number of objects and their size distribution. All the other models have higher number of objects (Fig. 14), mainly by generating more numerous objects with a size below about 10<sup>3</sup> km<sup>2</sup> (Fig. 15). On the other hand, the models (beside IFS) have a considerable low bias of objects having areas above 10<sup>4</sup> km<sup>2</sup>, i.e. scales matching mesoscale convective systems. The latter should be a robust finding, while the limitations in CCIC's horizontal resolution can play a role for areas below about 10<sup>3</sup> km<sup>2</sup>.

A rough view of the diurnal variation is found in Fig. 16. The CC<sub>n</sub> of CCIC averaged over the tropical region has a close to sinusoidal pattern, with maximum/minimum around local solar times of 20:00/8:00. This pattern is mainly driven by the diurnal cycle of FWP over land areas like the Amazonas, with peak values in FWP around LST 17:00, while the cycle over ocean areas is weaker and mean FWP peaks around sunrise (Leko, 2025). The diurnal cycle of IFS is very close to the one of CCIC, both in terms of amplitude and phase. Also ARPEGE, ICON, GFDL and GRIST show a cycle similar to CCIC. The other models must be considered to deviate in a significant manner. A closer analysis should make a separation between land and ocean. For example, the diurnal cycle of GEM has the characteristics of ocean areas, and it could be working well for this surface type.

| Dataset | $I_{ m org}$ | OIDRA | SCAI | MCAI  | ROME  | COP   | ABCOP |
|---------|--------------|-------|------|-------|-------|-------|-------|
| CCIC    | 0.80         | 0.47  | 0.67 | 0.991 | 108.5 | 0.026 | 36.9  |
| ARPEGE  | 0.80         | 0.45  | 2.50 | 3.676 | 30.9  | 0.014 | 84.3  |
| GEM     | 0.80         | 0.50  | 1.70 | 2.518 | 45.3  | 0.017 | 103.7 |
| GEOS    | 0.80         | 0.49  | 2.07 | 2.995 | 33.5  | 0.016 | 105.9 |
| GFDL    | 0.79         | 0.47  | 2.73 | 4.045 | 29.4  | 0.013 | 106.0 |
| GRIST   | 0.74         | 0.45  | 3.21 | 4.729 | 24.1  | 0.013 | 69.4  |
| GSAM    | 0.79         | 0.48  | 2.12 | 3.068 | 34.7  | 0.015 | 79.0  |
| IFS     | 0.78         | 0.48  | 0.73 | 1.075 | 95.5  | 0.029 | 26.9  |
| MPAS    | 0.84         | 0.45  | 1.38 | 2.011 | 58.0  | 0.016 | 101.0 |
| ICON    | 0.75         | 0.47  | 2.73 | 4.018 | 27.2  | 0.015 | 86.6  |

**Table 8.** Organization of convection according to seven measures. The results cover the CCIC satellite retrievals and nine GSRMs. Considered area and definition of objects as in Fig. 14.

#### 5.3 Convective organization indices

There is a growing interest in studying the organization and aggregation of convection in models and observations. In order to quantify and compare the organization of convection in different scenarios, numerous so-called organizational indices have been defined, which measure the level of organization. However, at this time, there exists no consensus on one such index. Mandorli and Stubenrauch (2024) reviewed and compared a number of such measures. In particular, they conclude that the use multiple indices is advantageous to successfully characterize the underlying ogranizational structure. Here, we apply seven of those indices on the GSRM and CCIC data, with results summarized in Table 8. Note that we modify the characteristic length scales used in Mandorli and Stubenrauch (2024) to allow for rectangular domains.

The first index,  $I_{\rm org}$ , is based solely on the distances between convective objects. It classifies the spatial organization of convective areas as random ( $I_{\rm org}=0.5$ ), clustered ( $I_{\rm org}>0.5$ ), or regular ( $I_{\rm org}

The SCAI and MCAI indices are based on the distances between objects, while MCAI additionally accounts for their size. Note that both indices reach maximum organization at zero; their lower bound. Mandorli and Stubenrauch (2024) shows that both indices strongly correlate with the number of convective objects. Our results reflect these findings (cf. Fig. 14). Accordingly, CCIC shows the strongest convective aggregation, with IFS in closest agreement among the GSRMs. Conversely, GRIST exhibits the weakest aggregation according to both indices.

The ROME index depends on the spatial proximity and size distribution of convective objects. Higher values indicate a greater degree of aggregation. In our data, ROME identifies IFS as the model with the strongest aggregation apart from the CCIC retrievals. On the other end, ICON exhibits the lowest aggregation. This ranking is consistent with Fig. 15, which reports much a higher number of large convective objects in CCIC and IFS compared to the other GSRMs. As such, our data supports the strong positive correlation with the objects mean size found in Mandorli and Stubenrauch (2024).

The COP index increases with the proximity and size of convective areas, with higher values indicating greater convective aggregation. However, Mandorli and Stubenrauch (2024) show that COP is not strongly correlated with the number, total area or mean size of the convective objects. According to this index, the convection in CCIC is more aggregated than in any of the GSRMs. Among the models, IFS exhibits the highest degree of convective aggregation and the closest agreement to CCIC, while GRIST and GFDL show the lowest levels aggregation.

Lastly, the ABCOP index is a modification of COP that has a strong correlation with the total area of convection over the studied domain (Mandorli and Stubenrauch, 2024). This index also reports IFS as the model that better agrees with CCIC, but also reports both of them as exhibiting lower aggregated convection than the rest of the models, in contrast with ROME, SCAI, MCAI and even COP.

In summary, the level of convective aggregation, and thus the ranking of both the GSRMs and CCIC depends on the chosen index. This reflects their different methodological approaches. SCAI, MCAI, ROME, and ABCOP consistently identify IFS as the closest match to CCIC in terms of convective aggregation. These indices exhibit strong correlation with the number and/or size of convective areas (Mandorli and Stubenrauch, 2024). As such, the results may be explained through Figs. 14 and 15. The three remaining indices COP,  $I_{org}$ , and OIDRA all only weakly correlate with the aforementioned variables. While COP also identifies IFS as the closest match to CCIC,  $I_{org}$  and OIDRA disagree in favour of other models.

## 645 6 Conclusions

650

Accurately representing atmospheric ice remains a major challenge for climate science, as remaining limitations have a critical influence on radiative transfer, vertical water transport, precipitation processes, and high cloud radiative feedback. This study examined the representation of atmospheric ice mass in various types of circulation models and satellite observations, with the frozen water path (FWP) as a common benchmark.

Global (Table 4) and zonal (Fig. 4) FWP means of considered CloudSat-based satellite retrievals (2C-ICE, DARDAR and AOP) are within about  $\pm 15$  % of the ensemble mean. This is a small spread compared to the poor agreement found in comparisons involving passive datasets (e.g. Duncan and Eriksson, 2018). However, the radar-based datasets can still share and

655

660

665

675

680

have counteracting biases, and an uncertainty in the order of 40 % can not be ruled out. Early results from some new sensors (EarthCARE and AWS) indicate that this more pessimistic view is in fact valid.

Global climate models remain biased low. Most CMIP6 and HighResMIP models substantially underestimate FWPs compared to satellite-based estimates, particularly in the tropics. The overall spread among models is reduced compared to earlier generations; however, the systematic bias persists. One could think that the underestimation was due to models not reporting all forms of ice in their output; in particular, many models do not include precipitating ice. However, Fig. 5 shows that those models that do include precipitating ice in their output do not have systematically higher FWP compared to those that do not.

GSRMs represent a significant step forward. They capture a more comprehensive hydrometeor spectrum and represent deep convection explicitly, resulting in more realistic FWP distributions. Although significant differences remain when compared to observations, particularly for extreme events and regional variability, they are in a relatively early stage of development and there is hope for further progress within this new modelling paradigm in the near future.

No consensus 'truth' exists. Both observations and models suffer from inherent limitations, leaving open the question of the true global distribution of atmospheric ice. This hinders the closure of the atmospheric water cycle and the quantification of associated radiative effects. This remark includes the fractioning between cloud ice, graupel, and snow; neither observations nor models provide a basis for distinguishing between the hydrometeor categories.

Future progress requires coordinated efforts. Advances in satellite missions (e.g., EarthCARE and ICI), retrieval methodologies, and GSRMs are essential. Cross-comparisons of independent datasets, alongside targeted process studies, will be crucial in reducing uncertainties. For the GSRMs, there should be efforts to pinpoint the sources of the ice mass disagreement. Microphysics schemes are a possible source, but other parametrisations are also relevant. Although GSRMs lack some of the problematic parametrisations of GCMs (convective parametrisation, cloud scheme), others remain, in particular the turbulence scheme, which was identified by Lang et al. (2023) as a major source of discrepancies in the representation of the moisture distribution.

To facilitate comparison with satellite retrievals, we recommend to extend the saved output from atmospheric models. In the case of GCMs, the first step would be to report complete ice masses (including all forms of precipitating ice) at all, with adding FWP as the initial step. This quantity is already covered in publicly available GSRM datasets, and the natural continuation is also to provide vertically resolved data. On this side, we recommend starting with the total mass, i.e., frozen water content (FWC), if data volumes make it impractical to keep cloud ice, snow, and graupel contents separated.

In summary, while significant progress was made since earlier assessments, atmospheric ice mass remains to be much less well constrained than other atmospheric variables. Narrowing the uncertainties around FWPs is crucial for improving confidence in climate projections, because it would help us better constrain high cloud radiative feedbacks.

Data availability. The CMIP6 models are available at aims2.llnl.gov. DYAMOND model output is available on the DKRZ. Chalmers Cloud Ice Climatology (CCIC) can be downloaded from https://registry.opendata.aws/ccic (Amell and Pfreundschuh, 2025). 2C-ICE data (Deng et al., 2025) were downloaded from the CloudSat Dataprocessing Cente. DARDAR-CLOUD v3-10 (Delanoë and Hogan, 2025) are available

at the ICARE Data and Services Center. The EarthCARE CPR\_CLD\_2A product dataset (ESA, 2025) was downloaded from the ESA Earth Online portal. The AWS FWP retrievals are based on AWS L1B data (EUMETSAT, 2025) distributed by EUMETSAT, and will be available at https://clouds-and-precip.group.

Author contributions. PE coordinated the study together with LI. HH, EM and MB processed and analysed the satellite retrievals. ABP and NM performed the same work on the model data. The text is written by PE, ABP, LI, SB and EM. All authors have been active in discussing the complete manuscript.

*Competing interests.* One of the co-authors is a member of the editorial board of Atmospheric Chemistry and Physics. The authors have no other competing interests to declare.

Acknowledgements. DYAMOND data management was provided by the German Climate Computing Center (DKRZ) and supported through the projects ESiWACE and ESiWACE2. The projects ESiWACE and ESiWACE2 have received funding from the European Union's Horizon 2020 research and innovation programme under grant agreements No 675191 and 823988. This work used resources of the Deutsches Klimarechenzentrum (DKRZ) granted by its Scientific Steering Committee (WLA) under project IDs bk1040 and bb1153.

The work by PE, HH and EM was in part funded by the Swedish National Space Agency (grants 2021-00077 and 2023-00139). This study is a contribution to the Strategic Research Area "ModElling the Regional and Global Earth system" (MERGE), with ABP fully funded by this programme. LI was supported by the Chalmers Gender Initiative for Excellence (Genie). For SB and MB, this work is a contribution to the Earth and Society Research Hub (ESRAH) at University of Hamburg, and to the Cluster of Excellence "CLICCS - Climate, Climatic Change, and Society".

The authors would like to thank Peter McEvoy for the processing of the AWS and CCIC zonal means presented in Fig. 5, and Adrià Amell for providing the CCIC data used in this study.

# 705 Appendix A: AOP radar inversions

This appendix complements Sec. 2.1.2 in the description of the ARTS onion peeling (AOP) algorithm. These radar inversions make use of pre-calculated look-up tables. The inversions are performed one layer at a time, beginning at the top of the atmosphere. The measured reflectivity in each layer is converted to an unattenuated reflectivity using the estimate of the two-way attenuation through the layer above. With the corrected value at hand, the local FWC and attenuation due to ice hydrometeors are derived from the look-up table matching the assumed particle habit and size distribution. The attenuation due to gases (nitrogen, oxygen and water vapour) is always considered. To include liquid water content (LWC) in the attenuation is optional. When considered, the microwave absorption of LWC is calculated following Ellison (2007). Amounts of water vapour and LWC are extracted from ERA5. The extinction of ice hydrometeors is included in all calculations in this study. However, AOP allows a rough scaling of this extinction to compensate for multiple scattering, based on results in Battaglia

**Figure B1.** Time series of monthly global means FWP during the period 1980-2014 for (a) CMIP6 historical simulations, and (b) CMIP6 HighRes hist-1950 simulations.

et al. (2010). In this work, the scaling factor is set to 0.5. If the attenuation in one layer is overestimated, it tends to result in a too high ice hydrometeor attenuation in lower layers as well. To avoid a runaway effect, the attenuation correction is not allowed to exceed 4 dB. As an additional measure, FWC is clipped at 5 g m<sup>-3</sup>. In the absence of a comprehensive database of scattering data for oriented particles, look-up tables are generated based on data for totally random particle orientation. The effect of particle orientation is instead included as a term subtracted from measured reflectivities, specified in dB.

# 720 Appendix B: Time series and trends of GCM FWPs

Time-series of monthly global mean FWP are shown in Fig. B1. Local changes in mean FWP are displayed in Figs. B2 and B3, as relative trends with respect to the local mean FWP over the time period considered.

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

**Figure B2.** Decadal FWP trends from 1980 to 2014 for the CMIP6 historical simulations. The trends based on the monthly mean values in each model grid cell for the full period. Red-shaded regions correspond to increases in FWP, whereas blue-shaded regions correspond to decreases in FWP.

**Figure B3.** Decadal FWP trends from 1980 to 2014 for the CMIP6 HighRes hist-1950 simulations. Red-shaded regions correspond to increases in FWP, whereas blue-shaded regions correspond to decreases in FWP.