# Peer review of "Advancements and continued challenges in global modelling and observations of atmospheric ice masses"

_EGUsphere, 2025_

## Referee Comment (RC1)

This manuscript by Eriksson et al presents a fascinating discussion of atmospheric ice mass in satellite retrievals, GCMs, and GSRMs. The paper is very thorough and reflects an impressive amount of work. It is well written and clearly organized, and the figures are effective.

The paper's greatest strength is the section focused on spaceborne remote sensing, which provided an intermediate-level overview of some of the many considerations that go into these retrievals. The authors' expertise was evident, and the level of detail is well calibrated for readers of ACP. I learned a lot from this section and, as the authors note, it fills a nice gap in the literature when it comes to comparison of these commonly used satellite products.

The GCM and GSRM sections were also strong, and the results in those sections will be of interest to the community. There were times when I was left hungry for more detail, but not every line of inquiry can be pursued in a single paper, and I think the authors triaged these questions effectively.

The majority of my comments below are suggestions or follow-up questions and do not necessarily require significant changes. Major comment #2 is probably the most significant and requires at least some additional detail. Otherwise, the paper seems publishable in its current form, and I defer to the authors to decide which comments are worth addressing in the paper.

-Adam Sokol

**Major Comments**

- 1. Section 4.2: The overview of regional trends in this section is relatively vague, and there are some conclusions presented that I don't think I would have reached myself. Nothing necessarily has to be changed, but my suggestions are:
  - o "Similar spatial patterns are observed in the decadal FWP trends of FGOALS-f3-L and FGOALS-f3-H"...while I appreciate the optimism, I'm just not sure Figure 8 bears this out. Tre is good agreement in the Arabian Sea and just south of the equator in the Atlantic. But the eastern tropical Pacific is the region with by far the most significant changes in Pfreundschuh et al (2025) and Figs B2/B3, and here FGOALS-f3-L shows a clear positive trend, while FGOALS-f3-H is mixed with a regional average that is probably close to zero. There is also significant disagreement north of the equator in the Atlantic and west of Australia. I think this figure could give different impressions to different readers, all of which might be reasonable, which is why I think some more specificity in the text could improve this section.
  - o Line 490-497: While I like the idea of distilling the GCMs into two simple groups, it just doesn't seem like the split is clear. I think it may be more accurate and interesting to comment on more specific features such as the fact that most, but not all, models predict a negative trend in the eastern Pacific ITCZ, which is in line with CCIC and PATMOS-x, but not ISCCP and ERA5, in Pfreundschuh et al (2025). Interestingly, FGOALS-f3-L predicts a positive trend here. I'm guessing intermodel differences are closely related to differences in regional SST
- 2. With regard to the CC\_n analysis and discussion of convective organization
  - o I have some reservations about the use of globally uniform FWP threshold to identify deep convection. Several factors may lead to differences in FWP within deep convection updraft intensity, depth of convection, surface temperature, etc. These things may systematically vary from region to region, meaning the use of a threshold based on the 97th percentile successfully identify convection in some regions but not in others. As a quick check, I looked at DARDAR v3 for the full 2008 year. In the eastern tropical Pacific (180W to 80W), the 97th FWP percentile

is ~0.4 kg/m2, while in the western Pacific (100-180E) it is 0.97 kg/m2. So, it seems likely that much of the convection in the E Pacific would go undetected when the global threshold is used. In Sokol & Hartmann (2020), we used a fixed FWP threshold to distinguish anvils and convective cores only over the very limited geographic regions from which the threshold was derived.

I realize that a similar argument could be made against any metric used to identify convection—none is perfect, but I think FWP introduces more complications than others. Using a different variable to identify convection would be a larger undertaking and probably does not make sense considering that this section does not have much to do with the rest of the paper. If the authors choose to retain this section, I think it is important to note the potential shortcomings of the FWP approach for identifying convection.

- o It seems like the CC\_n size/number is analysis is done for each day of output, and the daily results are used to generate the PDFs in Fig 14. Is the FWP threshold used to define convective cores the same for each day? Or does it reflect the 97th percentile of the FWP distribution just for that day? If the latter, I think this raises some additional complications, as even when the entire tropical belt is considered there may very different day-to-day statistics due to variability in the distribution and intensity of convection. Using a single metric for each model seems to make more sense, but does not fix the issues with regional differences described above.
- Putting aside the two points above, I am wondering about the results in Fig 14 not the analysis itself, but rather how reasonable it is to compare object size in the GSRMs to CCIC. It was shown in earlier sections that the convolutions in the CCIC algorithm reduce the product's resolution relative to the input data, and it can be seen clearly in Fig 1d that the width of convective towers can be exaggerated relative to the other retrievals. Could the spatial smearing have an impact on the size/number of the convective cores being shown in Fig 14? I don't have the technical expertise to make this judgement, but hopefully the authors can provide some insight as to whether this may be an issue.

**Minor comments (all optional)**

- 1. As discussed by the authors, the satellite retrievals assume that, below some temperature threshold, all condensate is ice as opposed to liquid. The authors mentioned differences in these thresholds between datasets and how suspected mixed-phase layers are treated. I'm not sure what to make of these different ways of dealing with mixed-phase clouds, namely, could these assumptions result in a significant overestimate of FWP, or in the authors' opinion are the simplifications all reasonable? We know that liquid can exist—even dominate—at temperatures well below the thresholds used by the retrievals (which from section 2.1.2 seem to vary from 0 to -7 C). Does this matter? Or is the mass of liquid that may mistakenly be classified as ice likely inconsequential, since ice particles tend to be larger and more massive? This was a question I was left with at the end of the paper and I would be interested to know the authors' thoughts given their expertise.
- 2. It may be interesting to mention in the introduction or conclusion efforts by the modeling community to avoid the arbitrary separation of ice into "cloud ice" and "precipitating ice" categories, namely the introduction of the P3 microphysics scheme which uses a single ice category to avoid these problematic distinctions (see Morrison and Milbrandt 2015; https://journals.ametsoc.org/view/journals/atsc/72/1/jas-d-14-0065.1.xml)

- 3. The text uses both g/m2 and kg/m2 for FWP. I suggest unifying all values to kg/m2 to match the figures, which makes the back-and-forth referencing to figures easier for the reader.
- 4. (optional) While there is significant attention devoted to the FWP occurrence fraction distributions, much of the paper is focused on *mean* FWP (be it global or zonal). This is reasonable of course, and I suspect most readers of this paper will agree that quantifying the total mass of atmospheric ice is intrinsically interesting. But it may be worthwhile noting in the conclusion that for many purposes—e.g., cloud radiative effect (CRE), in the case of my own interests—mean FWP is not the relevant metric. Berry & Mace 2014 (<a href="https://agupubs.onlinelibrary.wiley.com/doi/10.1002/2014JD021458">https://agupubs.onlinelibrary.wiley.com/doi/10.1002/2014JD021458</a>) have a very nice paper demonstrating that mean FWP does not tell you much about CRE, since mean FWP is heavily influenced by the FWP of convective cores but the thinner clouds are what largely determine CRE. So, while models may be very biased when it comes to mean FWP, they may still do a good job on CRE for the right reasons.

**Line Comments**

Line 35: By "not represented in their output", do you mean that it is not on the standard list of output variables for projects such as CMIP, or that these models are not built to output precipitating ice at all? Assuming the former, a clarification might be useful.

Line 39: GSMRs -> GSRMs

Line 38-40: I'm wondering what the authors mean when they say that ice microphysical processes have a direct physical representation in GSRMs. Clouds themselves are explicitly resolved, which is certainly a big improvement but microphysical processes are still parameterized.

Line 93: see also Figure 4 in Gasparini et al (2025) and Figure 1 in Sokol et al (2024). It could be worth mentioning the quite substantial differences between DARDAR v2 and v3, as shown in both of those figures in addition to the cited Atlas et al (2024)

Section 2.1.3- I think it would be helpful to include 1-2 sentences of what measurements exactly the CCIC product is ingesting in and what it is putting out. As is, the background on CCIC is scattered in a few places, and I am not exactly sure what my expectations for the product should be – i.e., is it most sensitive to small particles because it relies on passive IR measurements, most sensitive to larger particles because it is learning from CloudSat retrievals, or is the idea that it is doing both? Based on other ML-based retrievals, I think it is using merged, passive IR with overlapping CloudSat measurements to produce a 2C-ICE-like retrieval for every column of IR measurements. A sentence or two concisely describing this might be helpful.

Line 237: Did the authors confirm whether DARDAR indeed detects *no* cloud ice for these thin cirrus, or if the amount of ice detected just falls below the colorbar cutoff of 10^-5 kg/m3? These seems to be some ice detected by DARDAR that does not appear in Fig 1a—for example, Fig 1c shows nonzero DARDAR FWP at latitude~3.25 N, but in panel a these columns appear cloud-free.

Line 370: sensitive -> sensitivity

Section 3.2: I wonder if there would be better agreement between these DARDAR and 2C-ICE at low IWP if nighttime observations were used, when the lidar backscatter signal is easier to distinguish from background noise. If the authors agree, this might be a worthwhile point to mention.

Section 3.4: the unexpected agreement between DARDAR and AOP is quite striking. If the authors have any more speculation as to why AOP is so much closer to DARDAR than 2C-ICE, it could be interesting to include.

Section 3.7: the authors may also wish to mention IceCloudNet as another advancement in the ML-based retrieval category, although the recently published version is not global in coverage: <a href="https://journals.ametsoc.org/view/journals/aies/4/4/AIES-D-24-0098.1.xml">https://journals.ametsoc.org/view/journals/aies/4/4/AIES-D-24-0098.1.xml</a>

Line 470: I'm not sure what is meant by "none of the models can be falsified"

Line 471: Fig 6 shows that there is not much of a systematic difference in global mean FWP between models that include falling ice in their FWP and those that do not. It would be interesting to know if this is also the case for the zonal-mean FWP picture. While my gut tells me systematic differences are unlikely, they certainly might be plausible considering differences in governing processes between tropical and mid-latitude ice. No further analysis necessary, but if this can be easily done it might be interesting.

Line 540-541: the tropics are references earlier in the paragraph, but it might be good to reiterate here that this sentence applies only between  $\sim$ 20S and  $\sim$ 40N

Table 2, Table 6, Fig 12 lines 520, 543 (probably others that I've missed) – while many will make the connection, I recommend changing "GFDL" to "FV3" to match the name of this model in DYAMOND project

---

## Referee Comment (RC2)

The manuscript by Eriksson et al. addresses a fundamental challenge in quantifying atmospheric ice mass in satellite retrievals, GCMs, and GSRMs. It includes large a amount of results and summarises the main problems in both comparison between different observational products and their comparison with models. The paper is well written and organized but needs some changes before publishing.

Most of the following comments are rather questions to the authors to further explain their findings. However, several tables and figures are missing a full description and I recommend minor changes to the figures in order that the legends do not overlap with the plotted data. In some parts of the manuscript, I miss discussion/comparison of the results with the findings of others (see general and specific comments). Maybe the biggest issue I find is, to me, unclear satellite uncertainty. Do you find it as 10% as assumed in "satellite uncertainty" product (DARDAR +-10% is chosen), or 15% which you bring in the conclusions as you have showed through your results, or 40% based on the other paper? This needs to be clear and I find it very strange that the result from another paper (Austin et al. (2009)) is given in the abstract as an estimate for uncertainty. There are also some minor comments regarding parts which need to be clearer in order for the reader to fully understand the content. Finally, I have given some suggestions which do not require a lot of additional work but that I feel would improve our understanding on retrieval uncertainty and their comparison with simulations.

**General comments**

Abstract + Conclusions: While the abstract overall gives a beautiful picture of what to expect from the article, I cannot but notice how authors' estimation of retrieval uncertainty of 40% is misleading and not a result of the analysis performed. As mentioned later in the text, that uncertainty is reported in Austin et al. (2009) for FWC. In the conclusions, a 15% spread coming from the analysis is mentioned. I believe that this has to be clarified and reflected in the abstract and conclusions.

In several figures, the legends overlap with portions of plotted data. Please reposition the legends outside of the plotting area or where they do not obscure lines.

The captions under several tables (e.g. Table 3, Table 7) and figures do not clearly specify the geographic region and/or time period to which the results refer. Please revise the captions to provide this essential contextual information.

Given the length of the manuscript and the number of the results presented, it would also be helpful to restate the relevant time period in the main text when referring to tables or findings introduced in earlier sections. This would improve clarity for readers and reduce the need to look back through the manuscript for contextual information.

I would find it useful when describing own results to include information about others' findings. In some parts of the text, there is a nice comparison but on several occasions the text is written along the lines of "this has also been visualized/compared/analysed in..."

Satellite uncertainty in section 3.6, Figure 5 but also later leaves me with not being sure I understand what the grey area refers to (e.g. description in Figure 5: "The grey area represents the expected range according to older satellite observations (Sec. 3.6), based the zonal mean of DARDAR for July and August during 2007 to 2010)". Could you describe more clearly what you consider as satellite uncertainty? Is it the mean values during those 4 years and adding +-10% of the shaded area as described in Line 386? Also, does the DARDAR data here include only daytime or nighttime too? I

would also appreciate if the authors could add the mean value line. I was wondering if the assumed uncertainty is larger than the standard deviation of the DARDAR data for the years given? Moreover, since you are comparing DARDAR and 2C-ICE extensively, I would recommend to include and compare that DARDAR data (2007-2010) with 2C-ICE (2007-2010). It is also available on open source: TIWP and CIWP data used for accepted JGR paper 2020JD032848RR

Table 4 does not include information about the period (neither specific year or season – I gather it refers to the annual 2015 mean? Since you use DARDAR 2007-2010 as reference data (+-uncertainty), I think it would be useful to add the mean FWP for those years into the table, or later in the text, to understand overall multi-year variability of the data.

Generally, I am not sure what to expect from CCIC data but am a bit surprised with such large values since machine learning product uses passive remote sensing, especially in the tropics as I would expect that passive remote sensing will saturate fast (therefore, it is probably the result of ML statistics rather than retrieval?). I would appreciate if the authors could expand on it.

Section 4.2: The section is a bit unclear to me and maybe some conclusions are not very easy to reach. The manuscript is already long, but maybe adding some text to the appendix would help readers.

I find Figure 9 very interesting. I would be very much interested, if possible, to see what annual cycles other satellite data has (even for shorten time period).

General comment on GSRMs: Have you excluded the first 10 days of spin-up following Stevens et al. (2019), Lang et al., (2021), Corko et al. (2025 ...?

In my opinion, it would be preferable if the authors compared the model results with observations at least with respect to the season during which the DYAMOND model simulations were conducted. Despite not being nudged, models were initiated with ECMWF SSTs and therefore the convection is highly influenced with the period of simulations.

Section 5.2 (The tropical region): out of curiosity, do you know if convection is more active/stronger during northern summer than northern winter? I would think that this is the case also due to more land over the northern hemisphere. Would that impact the comparison of AC (%) and in particular CC (%) from the winter DYAMOND model and observations from Table 3 (please add to the captions the area and time period where missing)?

Could you explain the 99.99 percentile? Are those outliers (1 or 2 cases or more)? I am surprised that, in 2C-ICE, it goes to nearly 18kg/m² while with DARDAR to only 9.6 kg/m²? In the convective case you showed in Figure 1, both DARDAR and 2C-ICE go up to nearly 10 kg/m² in the convective core, as well as in figure 2. Can you show the scene of that 99.99 percentile case? Do you know what is so different in DARDAR for extreme cases compared not only to 2C-ICE but also to AOP nominal that constrains DARDAR FWP to cca 10 kg/m² maximum?

Line 93: "Comparisons of 2C-ICE and DARDAR are surprisingly few; exceptions include Deng et al. (2013); Winker et al. (2024); Atlas et al. (2024)." It would be great if you cited those papers and their results when you compare DARDAR and 2C-ICE (I feel there is no unique answer regarding the differences between the two products). In my experience, I have always felt they were "less different" than what you show here. Do you have an idea what could be a reason for such a

difference? Do you think it would be different if you analysed a different time period? For example, in Atlas et al. (2024), Figure3 shows that in winter period (February data from the years 2007–2012) FWP from DARDAR is substantially larger than in 2C-ICE (opposite to what you found in general (table 4/figure 4 etc...). I feel that generally, this kind of discussions is missing in some parts of the manuscript.

**Specific comments:**

- 1. Line 26-28: "It is noteworthy that the data request document for the latest Coupled Model Intercomparison Project (CMIP6) in effect defines cloud ice as the ice categories considered by the model's radiation scheme." I feel citation would be good.
- 2. Line 32-33: "Later studies comparing cloud ice from GCMs with various satellite observation include Eliasson et al. (2011); Li et al. (2012); Jiang et al. (2012); Komurcu et al. (2014); Li et al. (2020)." This is an example of only providing citations without any information. Could you provide their results or reason for their mention?
- 3.Line 77-78: "In any case, total ice is still the only mass quantity that has a clear definition." Could be expressed better.
- 4. Line 78-80: "DARDAR and 2C-ICE have been used as the reference in many studies, but are generally used separately." I think a few citations should be inserted after "many studies".
- 5. Line 81-82: "DARDAR and 2C-ICE do not offer sufficient coverage for addressing these questions and a dataset representing the state of the art for passive retrievals is also applied (denoted as CCIC)." CCIC has not been defined yet, so please define it. I understand from the text that it is a machine learning (ML) product based on passive observations. Therefore, I would not call it retrieval but how you mentioned ML product.
- 6. Line 102-103: "As older passive retrievals have been shown to have a strong bias with respect to CloudSat-based ones, they are excluded from this study." What do you consider as older passive remote sensing? I feel citations are missing.
- 7. Section 2.1.3. Passive dataset. Could you explain it a bit more? I am not sure what to expect from this kind of product. IF 2C-ICE has certain amount of ice connected to certain region and season, e.g. over the tropics where passive sensors saturate fast (and will retrieve much lower FWP), machine learning will indicate that there is too low FWP (compared to 2C-ICE) and artificially prescribe statistics from 2C-ICE despite non-physical retrieval relative to passive remote sensing? In that sense, which part is coming from retrieved values from passive remote sensing and which from "copying" the 2C-ICE retrieval? I think it would be helpful to add a few sentences about the product.
- 8. Line 190-191: I would replace "resolution" into a grid spacing. "Resolution of 5 km" means that the model grid length is a fraction of that (say, around 1 km or less). Therefore, when one refers to the actual model grid length, then "resolution" should be replaced with "grid length". "Resolution" can be used in a general sense, as in "low resolution models". One can also say "5 km grid" in place of "grid length of 5 km". I think there are a few places, also in the tables, where I feel it needs to be changed to grid spacing.
- 9. Figure 2: The information about the area of analysis is missing.

- 10. Section 3.3 and text related to Figure 3: To me, it appears that PDFs of FWP (Figure 3a) are very similar and do not vary a lot. In my understanding, everything under 5 g/m² can be detected only by LIDAR, which also loses its sensitivity for <1 or 2 g/m². Therefore, I am not sure if I would focus a lot on the FWP <  $10^{-3}$  kg/m². Also, I believe PDF (Figure3a y axis) should not have units. Regarding Figure 3b: could you please explain in more detail how you calculated it? I find values on y axis very low and very different than in the papers you have cited (e.g. Sokol and Hartmann (2020); Atlas et al. (2024).). I believe it is influenced by the number of bins?
- 11. Figure 5: Even though I appreciate the effort to use newly available data, I would still be very careful about using and interpreting EarthCARE data, as it is still experimental and in the validation process. As authors may be familiar with, there will be a few products available regarding the FWP and the product they use is just one of them. Therefore, at this stage, I would rather exclude this experimental data. However, in case the authors want to keep and show it, it has to be clear in the text, caption related to the figure and the legend, that this is still experimental data, yet to be verified, and instead of using "Earthcare", in the legend and elsewhere, the name of the product "CPR\_CLD\_2A" should be written because it could make an incorrect impression of EarthCARE data.
- 12. Line 442-443: "When those outliers were excluded, the remaining models showed differences of about a factor of 6 comparable to our findings." This result is surprising as it would turn out that there is no overall improvement in the model spread between old CMIP3 and new CMIP6 generations. This is the opposite of what you mention in the abstract. Could you please comment on this and include citations in the results' part that support the progress between the model generations?
- 13. Line 448: "Most models participating in the CMIP6 and CMIP6-HighRes underestimate the FWP when compared to satellite retrievals." In my opinion, this is clearly an understatement. Nearly all models in Figure 7 underestimate FWP (except one in each group and some small parts of 3-4 models are barely within the "huge" uncertainty of satellite observations). This might change once it is clear what you mean by satellite uncertainty (see my general comment)
- 14. Line 469-471: "Despite the large variability of FWP in between the models, we can still conclude that none of the models overestimate FWP compared to the satellite retrieval, and therefore none of the models can be falsified. However, it must be considered unlikely that non-reported ice masses can explain the gap to the satellite range for all models." I do not understand what the authors wanted to say here, e.g. "none of the models can be falsified"? Is it always "non-reported" ice as some GCMs include precipitating (snow) ice due to radiation scheme?
- 15. Figure 10(11) and related text: Since you are showing CCIC for February in Figure 10, why not include (calculate) its mean value and compare it with that value in DYAMOND models, instead of comparing mean FWP from models with annual CCIC and commenting it is about 3% lower than in February? That way, you would not need to speculate in the 1st paragraph in 5.1 section. In Figure 11, I appreciate the comparison with DARDAR for January to March in the years 2007-2010. Would you also consider adding CCIC zonal mean line from Figure 10? You are working with amazing data and interesting results which could, with minimum additional work, contribute to a very rich analysis (you could also calculate and comment the mean FWP for DARDAR 2007-2010 for January to March which is shown in Figure 11. As expected, it can already be seen in Figure 11 that, even though models mostly underestimate FWP, they are all in the range of DARDAR data when you compare it with the same winter season.

16. Line 544-546: "Model and observational distributions agree most closely at high occurrence fractions, where they exhibit similar peak values, while most models show higher fractions than the observations around FWP  $\approx 10^{-4}$  kg m-2." As I already commented before, you are describing FWP of 0.1 g/m², values which observations will most probably not retrieve, not even lidar? Figure 12: Again, I do not understand such small values on y axis. See also my comment 11.

17. Conclusion (Line 650-654): Here you mention that you showed around 15% spread, in the text for comparison with models, you assume DARDAR data as a reference with 10% spread, but in the abstract "estimate" uncertainty up to 40% (because what Austin et al 2009 found in their paper). And this 40% in the abstract, in my opinion, is misleading of what to expect from the paper since the spread you show is "much" lower. Also, mentioning that EarthCARE data could indicate the 40% uncertainty, based on preliminary, yet to be verified, one product would further mislead the reader and I find it not to be appropriate.

**Technical corrections:**

A minor point: in several places the manuscript does not clearly distinguish between satellites and the instruments they carry (e.g., CALIOP lidar on CALIPSO, the radar on CloudSat/EarthCARE). This appears to be a writing oversight. Please review the text to ensure that satellites and their respective instruments are correctly identified throughout.

Line 39: GSMRs should be GSRMs

Line 432: "However, we still compare the observations following Waliser et al. (2009), Jiang et al. (2012), Li et al. (2020), and others." – compare models/them with the observations

---

## Referee Comment (RC3)

**Comments on 'Advancements and continued challenges in global modelling and observations of atmospheric ice masses'**

This article reports an intercomparison of atmospheric ice mass, first between different retrievals, leading to an estimated uncertainty within the observations, and then between global climate model simulations as well as global storm-resolving model simulations in reference to these observations. The article is very rich in results. The comparison of atmospheric ice mass is not easy, as the range spans several orders of magnitude, and mean values over such a large range are not enough to fully understand differences. Therefore, distributions of atmospheric ice mass are also shown in observations and in global storm-resolved model simulations. In that way, four datasets, all based (more or less directly) on radar-lidar observations from space, and nine global storm-resolving model simulations are compared.

The comparison between the models is more reported than understood, in particular for the global climate models, but this reporting is a first step so that the different model teams can work on further improvement of the parameterizations.

As the outcome of this assessment is important, I recommend publishing, but only after a major revision. This revision is mostly to improve the structure of the article and to clarify certain points, so it should not be too difficult for the authors.

**Major comments**

**1.** The introduction explains well the problems in the definition of cloud ice. However, it is not completely clear if the final term 'frozen water path (FWP) corresponds to a grid-box average or an in-cloud average. From Pfreundschuh et al (2025) I deduce that it is the grid average. This should be clearly stated. The in-cloud FWP, which is directly retrieved by DARDAR and 2C-ICE, is also interesting to compare, as it is used in the cloud radiative transfer in the models.

**2.** The overarching goal of this article seems to be the assessment of the simulated atmospheric ice mass, but the authors also took the effort to intercompare four different datasets, based on satellite retrievals using radar-lidar or radar-only observations. In particular, they present results of the relatively new CCIC dataset, which is based on Machine Learning (ML) techniques trained on the CloudSat-lidar 2C-ICE product. So, the goal is actually two-fold. This should be more clearly formulated in the abstract and in the introduction.
Though several publications exist about this new dataset, it would be very helpful to clarify the description of this dataset and to show the uncertainty of this dataset which comes out of the applied Machine Learning technique, as explained in one of the earlier publications. More detailed questions and comments on this issue:

(a) In Pfreundschuh et al. (2025), the used ML technique for the CCIC is a convolutional neural network (CNN), while in this article the retrieval is given as quantile regression neural networks (QRNN). ***This is confusing.*** Indeed, the authors cite several articles which describe the retrieval, but ***it would help to give a more detailed overview of this retrieval***. I am very surprised how with only the use of one 11 micron brightness temperature (TB) together with the structure of the TB variability over regions of about 900 x 900 km2 (with 256 x 256 pixels) allows for such accurate prediction of IWP of the CloudSat-lidar 2C-ICE product, the latter given on a spatial resolution of about 1.5 x 2.5 km2. The TB depends on cloud height, on ice crystal habit and size distribution and on IWP. The TB also depends on season and daytime. How is this taken into account, in particular when the data are also expanded to other observational times than 1:30 AM and 1:30 PM LT?

(b) In general, regression neural networks give the right average compared to the dataset they are trained on, but scene-dependent biases exist when scenes with very large IWP are rare, as is the case in the tropics. This effect is even larger when the retrieved variable spreads out over several orders of magnitude. This reduces then the range of the ML-derived variable, as can be seen in Fig. 6 of Amell et al. (2025) or in Fig. 5 of Pfreundschuh et al. 2025. Somehow these biases show in the difference between the distributions in Figures 3. There is a large part with very small FWP, can the authors explain these cases?

(c) Indeed, the results are much better than those using only passive remote sensing, but it would be interesting to see the uncertainty of the ML retrieval. In Pfreundschuh et al. (2018), it is written that QRNNs also provide the uncertainty, but *I do not see this uncertainty quantified or presented in the current manuscript*.

**3.** The structure of the article:
(a) After section 2 (Data) which presents Satellite retrievals and models, it is confusing to see sections 3 Satellite retrievals, 4 GCMs and 5 GSRMs. I would include section 3 'Intercomparisons' and then put the initial sections 3-5 as subsections: 3.1 Satellite retrievals, 3.2 GCMs and 3.3 GSRMs.

(b) Furthermore, it is very confusing to see an outlook (section 3.7) in the middle of an article. Normally the outlook comes after the conclusion of a scientific article, which itself presents scientific results and their interpretation.

(c) The interpretation of Figure 5 needs some clarifications: The CCIC results are now shown for 10:30 and 22:30 LT, while they have been obtained via ML with a training at 1:30 and 13:30 LT. There is not one sentence on the reliability of this expansion in time. Also, what exactly is the satellite uncertainty shown in gray in Fig. 5? Another interesting point is that CCIC and SPARE-ICE show very similar zonal averages (except NH subtropics). Does this mean that the microwave information is useless in the retrieval of IWP (as CCIC only uses one IR channel)? Is it possible to give some explanations? Also, the authors state that the EarthCARE sensor and retrieval are improved. As the EarthCARE zonal mean is quite low in the tropics, does this mean that the high peaks in CCIC and SPARE-ICA and AWS are due to not-detection of thinner cirrus? This seems to be a huge effect.

(d) Many intercomparison results are shown, but for example to compare the global mean of a variable which spans several orders of magnitude is not a strong assessment. One interesting point here is that the IFS distribution (Fig. 11) does not agree with the observations, but the near-global mean does!
Since the intercomparison sections are quite long, one could probably take the comparison of the global means to the supplementary material and include the global mean values to Table 1 which could also be moved to a supplement, and then one starts this section with the comparison of zonal means. The same for the global means of the GSRMs: I suggest combining Table 6 with Table 2 and moving them to the supplementary material.

**4.** Retrieval trueness and estimated uncertainty in section 3.6:
(a) I have difficulties to follow the argumentation. From Fig. 1 it looks like 2C-ICE seems to be more sensitive to thin Cirrus and therefore the distribution in Fig. 3 shows two peaks. 2C-ICE also seems to have a larger range in FWP towards larger FWP. Since the range towards the larger FWP counts more in the mean than the larger range towards smaller FWP, the authors find a 24% larger mean. Why should you put more weight on DARDAR and AOP, the latter only using CloudSat data?

(b) The uncertainty range of 40% is assumed without any further explanation, and this is highlighted as result in the abstract. Why do you not show the uncertainty of CCIC which you claim in earlier articles can be obtained via QRNN? The sensitivity studies in section 3.5 show another part of uncertainty, based on the microphysical assumptions. You could base your argumentation on these findings.

**Minor comments**

Title: 'ice mass' instead of 'ice masses'?, same in line 11

p 1, l 6 -7: 'but its accuracy is limited by biases inherited from its training dataset' : it is true that ML can as best be the same as the training dataset and therefore naturally includes its biases. However, this is trivial, and I would like to see in the abstract also mentioned the additional biases and uncertainty linked to the reduced input.

p 4, l 93: you may add Vidot et al. 2015 (DOI: 10.1002/2015JD023462), they compared IWC profiles for small and large COD (Fig. 4).

Section 2.1.2: I would move the second paragraph (p 5, l 120-123) to the front of this section

p 4, l 116: take out 'retrieval'

p 5, l 143: 'radar bin' perhaps 'radar vertical segment' ? is each bin or vertical segment about 0.5 km ?

p 7, l 6 & 7: please add 'boreal' in front of 'Summer' and 'Winter'

p 7, l 210-211: we sum up … (IWP, GWP, SWP): is this weighted by their fraction within the grid ?

p 19, l 434-435 Section 4: 'the overall assessment is based on global means':
This is really a pity, but probably CMIP6 results only provide the monthly means? It would be important to add in the conclusions that distributions should be added as output for CMIP7. However, you need also to mention that the distributions in Figures 3 may change their shape when reducing the spatial resolution to 100 or 250 km. Did you have a look how they would change?

Comparison of zonal means (Fig. 7): the authors compare grid averages of FWP from the model simulations to the range in satellite observations coming from nadir tracks; how do the authors build grid averages if there is only a narrow track within a grid of the GCM spatial resolution? Here, actually the CCIC dataset may be useful as it is expanded to fill a whole grid, even though additional uncertainty is added due to ML expansion.

Section 5: Why do you limit the GSRM means to 60N-60S while the GCMs are averaged over 90N-90S?

Figure 10: instead of (or in addition to) comparing the mean FWP of CCIC and the 9 models, one could show the difference map between both estimations in order to see where there may be differences.

Figure 16: It is known that the diurnal cycle of convection differs over ocean and over land, therefore a comparison seems only to make sense when ocean and land are separated.

*Section 5.3:*
*It is interesting that the authors also explore convective indices, but it is difficult to follow this section.*
p 31, l 604-605: 'In particular, they conclude that the use of multiple indices is advantageous to successfully characterize the underlying organizational structure. '
For me, it looks like they concluded first that several of these indices don't fulfil certain quality criteria, like sensitivity to noise under certain conditions, to spatial resolution etc. and this can explain

differences in conclusions about convective organization when using different indices; and second that these indices may not be enough to completely characterize organization. Another conclusion was that some indices are highly correlated with one simple variable, like ABCOP reflects the total area of convective objects, while ROME is very strongly correlated with the mean size of the objects. The latter can be seen by comparing Fig. 15 with Table 8. SCAI and MSCAI strongly depend on the number of objects, which may be very noisy. Since Iorg does not consider the size of the objects, the conclusion on organization using Iorg and ROME does not agree. Perhaps one can add in the table the correlations with the corresponding variable, and it would also be good to highlight in bold or italic the largest and smallest for each index.

Conclusions:
According to the questions and comments before, some parts need to be rewritten, in particular the 2. paragraph p 32, l 652 – p33, l 654).
p 33, l 655: 'Global climate models remain biased low': it may be good to add here that this is on grid average; it can be different in in-cloud IWP (see above).

p 35, 653-654: I do not understand the last sentence of the paragraph. Why indicate preliminary results from new sensors, for which the retrieval is also based on assumptions indicate the pessimistic view?

p 35, l 721-722: I do not see any Figures B2 and B3 in the newest manuscript.

**Typos**

p 2, l 32: 'observations' instead of 'observation'

p 2, l 39: 'GSRMs' instead of 'GSMRs'

p 3, l 67: take out 'the' in 'the the'

p 5, l 144: 'a unit' instead of 'an unit'

p 14, legend of Table 3, last line: 'it is stressed' instead of 'it stressed'

p 14, l 313: 'two datasets' instead of 'two dataset'

p 17, l 386: 'DARDAR instead of 'DARADR'

p 18, l 413: add 'to' between 'up 664 GHz'

p 19, legend of Fig. 5: 'based on the zonal mean' instead of 'based the zonal mean'

p 26, l 552: change the first 'between' to 'in' would be easier to understand

p 26, l 553: 'latter' instead of 'later'

p 31, l 605: 'organizational' instead of 'ogrnizational'

p 33, l 685: in 'Data availability': 'Center' instead of 'Cente'

---

## Author Comment (AC2)

**Response to review by Adam Sokol**

*This manuscript by Eriksson et al presents a fascinating discussion of atmospheric ice mass in satellite retrievals, GCMs, and GSRMs. The paper is very thorough and reflects an impressive amount of work. It is well written and clearly organized, and the figures are effective. The paper's greatest strength is the section focused on spaceborne remote sensing, which provided an intermediate-level overview of some of the many considerations that go into these retrievals. The authors' expertise was evident, and the level of detail is well calibrated for readers of ACP. I learned a lot from this section and, as the authors note, it fills a nice gap in the literature when it comes to comparison of these commonly used satellite products.*

*The GCM and GSRM sections were also strong, and the results in those sections will be of interest to the community. There were times when I was left hungry for more detail, but not every line of inquiry can be pursued in a single paper, and I think the authors triaged these questions effectively.*

*The majority of my comments below are suggestions or follow-up questions and do not necessarily require significant changes. Major comment #2 is probably the most significant and requires at least some additional detail. Otherwise, the paper seems publishable in its current form, and I defer to the authors to decide which comments are worth addressing in the paper.*

- We thank the reviewer for noticing the amount of work that is behind the study, finding the efforts worthwhile and understanding the aims and practical constraints of the manuscript. It is also appreciated that the review is clear about what are strong suggestions and what we can consider as optional.

- Please, find below replies to your comments. We gave emphasis to points where there are commonalities with the input from one or both of the other referees, still carefully considering all points raised.

**Major Comments**

*1. Section 4.2: The overview of regional trends in this section is relatively vague, and there are some conclusions presented that I don't think I would have reached myself. Nothing necessarily has to be changed, but my suggestions are:*

- *"Similar spatial patterns are observed in the decadal FWP trends of FGOALS-f3-L and FGOALS-f3-H"…while I appreciate the optimism, I'm just not sure Figure 8 bears this out. There is good agreement in the Arabian Sea and just south of the equator in the Atlantic. But the eastern tropical Pacific is the region with by far the most significant changes in Pfreundschuh et al (2025) and Figs B2/B3, and here FGOALS-f3-L shows a clear positive trend, while FGOALS-f3-H is mixed with a regional average that is probably close to zero. There is also*

> *significant disagreement north of the equator in the Atlantic and west of Australia. I think this figure could give different impressions to different readers, all of which might be reasonable, which is why I think some more specificity in the text could improve this section.*

- We agree that Figure 8 can give different impressions to different readers. After re-evaluating this part of the manuscript we concluded that Figure 8 does not provide additional novel information and therefore we decided to remove it.

  > - *Line 490-497: While I like the idea of distilling the GCMs into two simple groups, it just doesn't seem like the split is clear. I think it may be more accurate and interesting to comment on more specific features – such as the fact that most, but not all, models predict a negative trend in the eastern Pacific ITCZ, which is in line with CCIC and PATMOS-x, but not ISCCP and ERA5, in Pfreundschuh et al (2025). Interestingly, FGOALS-f3-L predicts a positive trend here. I'm guessing intermodel differences are closely related to differences in regional SST*

- Following the reviewer's recommendation, we have rewritten the text.

  > *2. With regard to the CC_n analysis and discussion of convective organization*

  > - *I have some reservations about the use of globally uniform FWP threshold to identify deep convection. Several factors may lead to differences in FWP within deep convection – updraft intensity, depth of convection, surface temperature, etc. These things may systematically vary from region to region, meaning the use of a threshold based on the 97th percentile successfully identify convection in some regions but not in others. As a quick check, I looked at DARDAR v3 for the full 2008 year. In the eastern tropical Pacific (180W to 80W), the 97th FWP percentile is ~0.4 kg/m2, while in the western Pacific (100-180E) it is 0.97 kg/m2. So, it seems likely that much of the convection in the E Pacific would go undetected when the global threshold is used. In Sokol & Hartmann (2020), we used a fixed FWP threshold to distinguish anvils and convective cores only over the very limited geographic regions from which the threshold was derived. I realize that a similar argument could be made against any metric used to identify convection — none is perfect, but I think FWP introduces more complications than others. Using a different variable to identify convection would be a larger undertaking and probably does not make sense considering that this section does not have much to do with the rest of the paper. If the authors choose to retain this section, I think it is important to note the potential shortcomings of the FWP approach for identifying convection.*

- We understand the concern. First of all, we have added comments to the text to remind the reader that we analyze high FWP regions, and not convection directly.

- It is correct that the highest FWP values tend to be localised to some regions, but it should still be reasonable to assume that FWP and the convective strength are related, and we are then hopefully looking at the areas of highest convection both with CCIC and inside the models. In fact, we relax the test of the models by ignoring exactly where inside the tropics the highest FWP values occur. That said, as shown by the figure below, the spatial patterns of CC_n occurrence fractions are similar between the models, and in agreement with CCIC. The figure also shows that several areas have significant portions of CC_n. The coloured lines indicate the region we use in Secs. 5.2 and 5.3, and for which the FWP threshold has been derived (individually for each dataset). The found threshold has, for this figure, also been applied outside of the tropics, for context.

[Figure]

- *It seems like the CC_n size/number is analysis is done for each day of output, and the daily results are used to generate the PDFs in Fig 14. Is the FWP threshold used to define convective cores the same for each day? Or does it reflect the 97th percentile of the FWP distribution just for that day? If the latter, I think this raises some additional complications, as even when the entire tropical belt is considered there may very different day-to-day statistics due to variability in the distribution and intensity of convection. Using a single metric for each model seems to make more sense, but does not fix the issues with regional differences described above.*

- We use the later, but see why it could be misunderstood. We now write: "with a common threshold for the complete time period".

  - *Putting aside the two points above, I am wondering about the results in Fig 14 – not the analysis itself, but rather how reasonable it is to compare object size in the GSRMs to CCIC. It was shown in earlier sections that the convolutions in the CCIC algorithm reduce the product's resolution relative to the input data, and it can be seen clearly in Fig 1d that the width of convective towers can be exaggerated relative to the other retrievals. Could the spatial smearing have an impact on the size/number of the convective cores being shown in Fig 14? I don't have the technical expertise to make this judgement, but hopefully the authors can provide some insight as to whether this may be an issue.*

- In the discussion of Fig. 15, we wrote "while the limitations in CCIC's horizontal resolution can play a role
for areas below about 1000 km².", but it is correct that the limited horizontal resolution shall be considered already for Fig. 14. We have changed the text accordingly, and put significantly more emphasis on the issue.

**Minor comments (all optional)**

- *1. As discussed by the authors, the satellite retrievals assume that, below some temperature threshold, all condensate is ice as opposed to liquid. The authors mentioned differences in these thresholds between datasets and how suspected mixed-phase layers are treated. I'm not sure what to make of these different ways of dealing with mixed-phase clouds, namely, could these assumptions result in a significant overestimate of FWP, or in the authors' opinion are the simplifications all reasonable? We know that liquid can exist—even dominate —at temperatures well below the thresholds used by the retrievals (which from section 2.1.2 seem to vary from 0 to -7 C). Does this matter? Or is the mass of liquid that may mistakenly be classified as ice likely inconsequential, since ice particles tend to be larger and more massive? This was a question I was left*

> *with at the end of the paper and I would be interested to know the authors'*
> *thoughts given their expertise.*

- We thank the reviewer for the trust in our expertise, but want to clarify that we are developing passive retrievals and our knowledge on radar observations is not as deep. There were some clues in the text. In Sec. 2.1.2 we wrote "Cloud liquid droplets are too small to generate significant back-scattering at 94 GHz, but the DARDAR approach still assumes that there exist no super-cooled liquid droplets with a size matching drizzle or stronger rain." As Table 5 shows, ignoring the liquid cloud water gives an underestimation (as the attenuation is underestimated). Assuming that all back-scattering below 0C should be incorrect, a fraction should be due to larger liquid droplets. That is, our assumption here is that DARDAR makes an overestimation of FWP, but we feel uncertain about the magnitude. The test in Table 5, setting the temperature threshold at -5C, should be a worst case estimate. In any case, we have here two effects impacting the retrievals with different signs, and the net effect could be close to zero (for large scale averages, not single retrievals).
We think it is would be distracting to discuss these issues in detail in the text. However, in our rewriting of Sec. 3.6, we found it suitable to make some comments in this direction.

> *2. It may be interesting to mention in the introduction or conclusion efforts by the*
> *modeling community to avoid the arbitrary separation of ice into "cloud ice"*
> *and "precipitating ice" categories, namely the introduction of the P3*
> *microphysics scheme which uses a single ice category to avoid these*
> *problematic distinctions (see Morrison and Milbrandt 2015; https://journals.am*
> *etsoc.org/view/journals/atsc/72/1/jas-d-14-0065.1.xml)*

- Thanks for the input. It made us spot a resemblance with some retrievals in development. We certainly agree that this is a good way forward, and now bring this up in the Conclusions.

> *3. The text uses both g/m2 and kg/m2 for FWP. I suggest unifying all values to*
> *kg/m2 to match the figures, which makes the back-and-forth referencing to*
> *figures easier for the reader.*

- Suggestion adopted.

> *4. (optional) While there is significant attention devoted to the FWP occurrence*
> *fraction distributions, much of the paper is focused on mean FWP (be it global*
> *or zonal). This is reasonable of course, and I suspect most readers of this paper*
> *will agree that quantifying the total mass of atmospheric ice is intrinsically*

> *interesting. But it may be worthwhile noting in the conclusion that for many purposes—e.g., cloud radiative effect (CRE), in the case of my own interests— mean FWP is not the relevant metric. Berry & Mace 2014 ([https://agupubs.onlin elibrary.wiley.com/doi/10.1002/2014JD021458](https://agupubs.onlinelibrary.wiley.com/doi/10.1002/2014JD021458)) have a very nice paper demonstrating that mean FWP does not tell you much about CRE, since mean FWP is heavily influenced by the FWP of convective cores but the thinner clouds are what largely determine CRE. So, while models may be very biased when it comes to mean FWP, they may still do a good job on CRE for the right reasons.*

- This is correct, the mean FWP is not very relevant for CRE, but its distribution has some relevance (see citation below). However, we see a broader motivation. We wrote (towards the end of the Introduction): "We focus the comparisons on FWP for several reasons. FWP is directly proportional to the latent heat released during the conversion of water vapour to ice. It can be used to estimate precipitation efficiencies, the fraction of hydrometeor condensation that ends up as surface precipitation (Kukulies et al., 2024). The impact on longwave radiation at the top of the atmosphere is, on average, proportional to the logarithm of FWP (Deutloff et al., 2025)."
  To be even clearer, we have now incorporated Berry and Mace (2014) in this text, has added a citation to put more emphasis on the importance of latent heating and another one referring to tracking of convective storms.

**Line Comments**

> *Line 35: By "not represented in their output", do you mean that it is not on the standard list of output variables for projects such as CMIP, or that these models are not built to output precipitating ice at all? Assuming the former, a clarification might be useful.*

- The IPCC list on Standard Output from Coupled Ocean-Atmosphere GCMs (see [https://pcmdi.llnl.gov/ipcc/standard_output.html](https://pcmdi.llnl.gov/ipcc/standard_output.html)) includes the following variables that are relevant in the context of this study: atmosphere_cloud_ice_content (clivi), precipitation_flux (pr) and convective_precipitation_flux (prc), the latter both including liquid and solid phase. There is no specific standard variable that can be used to estimate the precipitating ice hydrometeors. Besides, many models do remove precipitating ice crystals within one time step, which makes accounting for that contribution to the ice mass difficult.

> *Line 39: GSMRs -> GSRMs*

- Changed.

*Line 38-40: I'm wondering what the authors mean when they say that ice microphysical processes have a direct physical representation in GSRMs. Clouds themselves are explicitly resolved, which is certainly a big improvement but microphysical processes are still parameterized.*

- Changed.

*Line 93: see also Figure 4 in Gasparini et al (2025) and Figure 1 in Sokol et al (2024). It could be worth mentioning the quite substantial differences between DARDAR v2 and v3, as shown in both of those figures in addition to the cited Atlas et al (2024)*

- A good point. The text has been changed to incorporate this information.

*Section 2.1.3– I think it would be helpful to include 1-2 sentences of what measurements exactly the CCIC product is ingesting in and what it is putting out. As is, the background on CCIC is scattered in a few places, and I am not exactly sure what my expectations for the product should be – i.e., is it most sensitive to small particles because it relies on passive IR measurements, most sensitive to larger particles because it is learning from CloudSat retrievals, or is the idea that it is doing both? Based on other ML-based retrievals, I think it is using merged, passive IR with overlapping CloudSat measurements to produce a 2C-ICE-like retrieval for every column of IR measurements. A sentence or two concisely describing this might be helpful.*

- Sec 2.1.3 has been expanded to provide more information on CCIC. However, to not introduce a poor balance with respect to the other retrievals, we still try to keep the description of CCIC compact. That is, we don't want the CCIC to overshadow DARDAR and 2C-ICE. See also our reply to referee #3.

*Line 237: Did the authors confirm whether DARDAR indeed detects no cloud ice for these thin cirrus, or if the amount of ice detected just falls below the colorbar cutoff of 10^-5 kg/m3? These seems to be some ice detected by DARDAR that does not appear in Fig 1a—for example, Fig 1c shows nonzero DARDAR FWP at latitude~3.25 N, but in panel a these columns appear cloud-free.*

- We have looked at this. It is not a question of the colourbar cutoff, it's the resolution of the plot. There is indeed a tiny amount of ice in DARDAR at latitude ~3.3 N, that extends slightly under 500m in height but only 0.1 degrees in latitude. Since we plot over quite a wide range of latitudes, this pixel-wide area of IWC doesn't appear in Fig. 1a. However, since we use a line plot in Fig. 1c, it appears. However, there are very few of these pixel-wide regions of IWC (only around 5 of them), so the statement that DARDAR detects no cloud ice for these thin cirrus is still valid.

*Line 370: sensitive –> sensitivity*

- Changed.

*Section 3.2: I wonder if there would be better agreement between these DARDAR and 2C-ICE at low IWP if nighttime observations were used, when the lidar backscatter signal is easier to distinguish from background noise. If the authors agree, this might be a worthwhile point to mention.*

- It is a valid point, that has been added to the text.

*Section 3.4: the unexpected agreement between DARDAR and AOP is quite striking. If the authors have any more speculation as to why AOP is so much closer to DARDAR than 2C-ICE, it could be interesting to include.*

- The closer agreement to DARDAR can partly be explained by the common 0C threshold, a similarity that is now explicitly mentioned, but we can still not explain why their zonal means end up to be so similar. We are surprised and remind that individual AOP and DARDAR retrievals can disagree significantly (see Figs 1-3).

*Section 3.7: the authors may also wish to mention IceCloudNet as another advancement in the ML- based retrieval category, although the recently published version is not global in coverage: https://journals.ametsoc.org/view/journals/aies/4/4/AIES-D-24-0098.1.xml*

- This article was published after our submission. A sentence referring to IceCloudNet has now been added.
- We use IceCloudNet to exemplify the trend of increased usage of machine learning (ML). When looking at the manuscript with fresh eyes, we felt a need to give a more balanced view on ML. For this reason we have added text to the Conclusions, with a critical discussion of the uncertainty estimates provided. There is here room for improvements in "traditional" retrievals, but, unfortunately, ML in general constitutes a step backwards when it comes to error reporting and comparison to other retrievals (beside the data used for training).

*Line 470: I'm not sure what is meant by "none of the models can be falsified"*

- Here we tried to be very formal, pointing out that there can be non-reported ice masses (even when including "snow" in IWP) that theoretically could move the models up to the satellite range, but this is very unlikely. Anyhow, the topic is in fact discussed elsewhere. Accordingly, we have simply removed the paragraph, in order to not cause confusion and make the manuscript a bit shorter.

> *Line 471: Fig 6 shows that there is not much of a systematic difference in global mean FWP between models that include falling ice in their FWP and those that do not. It would be interesting to know if this is also the case for the zonal-mean FWP picture. While my gut tells me systematic differences are unlikely, they certainly might be plausible considering differences in governing processes between tropical and mid-latitude ice. No further analysis necessary, but if this can be easily done it might be interesting.*

- It is the same in the zonal-mean FWP picture - there is no systematic difference between the models including falling ice or not. There is not much of a latitudinal pattern either. The models including falling ice that have a low global mean FWP (for example the CMCC, EC-Earth, NorESM and IFS model family) have also overall lower zonal means over all latitudes. The FGOALS model that have a very high global mean FWP show also higher zonal means over all latitudes that are in the range of the satellite observations. One exception is the model KACE-1-0-G which has a comparable high global mean FWP which results from a high latitudinal FWP at the poles (the FWP value at the equator is comparable low). We added this additional information to the manuscript: "Again, no systematic difference between the models including falling ice or not is found and for most models a small relative latitudinal difference in low/high FWP values can be seen, i.e. a model with low quasi-global monthly FWP means also shows low zomal mean monthly FWP values over the whole latitude range.".

> *Line 540-541: the tropics are references earlier in the paragraph, but it might be good to reiterate here that this sentence applies only between ~20S and ~40N*

- The final comments in the paragraph in fact refer to all latitudes. To make this clearer, this summary part is now placed in a separate paragraph. We also removed a comparison to the satellite observations, as it also could add to the confusion.

> *Table 2, Table 6, Fig 12 lines 520, 543 (probably others that I've missed) – while many will make the connection, I recommend changing "GFDL" to "FV3" to match the name of this model in DYAMOND project*

- Renaming done. Beside using FV3, GSAM has been changed to SAM in the same spirit.

---

## Author Comment (AC3)

**Response to review by Karol Ćorko**

> *The manuscript by Eriksson et al. addresses a fundamental challenge in quantifying atmospheric ice mass in satellite retrievals, GCMs, and GSRMs. It includes large a amount of results and summarises the main problems in both comparison between different observational products and their comparison with models. The paper is well written and organized but needs some changes before publishing.*
>
> *Most of the following comments are rather questions to the authors to further explain their findings. However, several tables and figures are missing a full description and I recommend minor changes to the figures in order that the legends do not overlap with the plotted data. In some parts of the manuscript, I miss discussion/comparison of the results with the findings of others (see general and specific comments). Maybe the biggest issue I find is, to me, unclear satellite uncertainty. Do you find it as 10% as assumed in "satellite uncertainty" product (DARDAR +−10% is chosen), or 15% which you bring in the conclusions as you have showed through your results, or 40% based on the other paper? This needs to be clear and I find it very strange that the result from another paper (Austin et al. (2009)) is given in the abstract as an estimate for uncertainty. There are also some minor comments regarding parts which need to be clearer in order for the reader to fully understand the content. Finally, I have given some suggestions which do not require a lot of additional work but that I feel would improve our understanding on retrieval uncertainty and their comparison with simulations.*

- We thank for finding our manuscript interesting and well written, and providing clear suggestions for improvements. Yes, in some parts we are brief, deliberately, in order to keep manuscript's length down. In our revision, we have added information, but still have considered the overall length of the manuscript. It is clear that we failed in describing the logic and definition of our "satellite uncertainty" used in the later sections. We have rewritten that part totally and are now not using the easy way out of referring to Austin et al.

- Please, find below replies to your comments. We gave emphasis to points where there are commonalities with the input from one or both of the other referees, still carefully considering all points raised.

**General comments**

> *Abstract + Conclusions: While the abstract overall gives a beautiful picture of what to expect from the article, I cannot but notice how authors' estimation of retrieval uncertainty of 40% is misleading and not a result of the analysis performed. As mentioned later in the text, that uncertainty is reported in Austin et al. (2009) for*

*FWC. In the conclusions, a 15% spread coming from the analysis is mentioned. I believe that this has to be clarified and reflected in the abstract and conclusions.*

- We have made several changes around this. The first part of the abstract has been rewritten. The section "3.6 Retrieval trueness" is totally rewritten, including a new discussion around the results in Sec 3.5 leading to 30% uncertainty (instead of 40%). A reference to Sec 3.6 has been added in the Conclusion, for increased clarity.

*In several figures, the legends overlap with portions of plotted data. Please reposition the legends outside of the plotting area or where they do not obscure lines.*

- We have changed figures to remove overlaps. We made no change to Fig. 13 as we think the overlap here does not cause any confusion at all.

*The captions under several tables (e.g. Table 3, Table 7) and figures do not clearly specify the geographic region and/or time period to which the results refer. Please revise the captions to provide this essential contextual information.*

- Information has been added.

*Given the length of the manuscript and the number of the results presented, it would also be helpful to restate the relevant time period in the main text when referring to tables or findings introduced in earlier sections. This would improve clarity for readers and reduce the need to look back through the manuscript for contextual information.*

- In general, we assume that the variability between time periods is considerable smaller than the deviations between datasets, and it is not critical to have the exact time period actively in memory when reading the text. However, this is not obvious in all parts. We should have been more clear when comparing results in Sec. 5 (for 24h, Feb 2020) with values from Sec. 3 (for 1:30, 2015). Comments and discussion have been added to Sec. 5. For example, CCIC has been added to both Table 6 and 7, clarifying that the statistical measures are quite similar despite the different time coverages.

*I would find it useful when describing own results to include information about others' findings. In some parts of the text, there is a nice comparison but on several occasions the text is written along the lines of "this has also been visualized/compared/analysed in…"*

- We hope the changes in response to all three referees have fixed these issues, at least the worst ones.

*Satellite uncertainty in section 3.6, Figure 5 but also later leaves me with not being sure I understand what the grey area refers to (e.g. description in Figure 5: "The grey area represents the expected range according to older satellite observations (Sec. 3.6), based the zonal mean of DARDAR for July and August during 2007 to 2010)". Could you describe more clearly what you consider as satellite uncertainty? Is it the mean values during those 4 years and adding +-10% of the shaded area as described in Line 386? Also, does the DARDAR data here include only daytime or nighttime too? I would also appreciate if the authors could add the mean value line. I was wondering if the assumed uncertainty is larger than the standard deviation of the DARDAR data for the years given?*

- As mentioned, we have rewritten Sec 3.6 from scratch, including to make a judgment more clearly based on Sec 3.5 (that was the idea from the start). We hope that the new text clarifies our approach. We prefer to not add the center of uncertainty ranges so to not clutter the figures, which in some cases already contain many lines.

*Moreover, since you are comparing DARDAR and 2C-ICE extensively, I would recommend to include and compare that DARDAR data (2007-2010) with 2C-ICE (2007-2010). It is also available on open source: TIWP and CIWP data used for accepted JGR paper 2020JD032848RR*

- The logic behind the comment seem to be to check if differences between DARDAR and 2C-ICE vary between time periods. This is of course a relevant question, that we had considered but failed to discuss in the text. Based on results in Pfreundschuh et al. (2025), we argue that the results derived for 2015 are valid for the complete CloudSat era. Text has been added to Sec. 4.3 to clarify this, and this is also pointed out at the start of Sec. 3.

*Table 4 does not include information about the period (neither specific year or season – I gather it refers to the annual 2015 mean? Since you use DARDAR 2007-2010 as reference data (+-uncertainty), I think it would be useful to add the mean FWP for those years into the table, or later in the text, to understand overall multi-year variability of the data.*

- The year has been added to the text of Table 4. We refer to the answer above for not expanding the table. To be clear, the logic is that in Secs. 3.1-3.6 we are only using data from 2015 (as we had ready collocations for that period). It is just for determining the uncertainty range in later parts we use 2007-10 means.

> *Generally, I am not sure what to expect from CCIC data but am a bit surprised with such large values since machine learning product uses passive remote sensing, especially in the tropics as I would expect that passive remote sensing will saturate fast (therefore, it is probably the result of ML statistics rather than retrieval?). I would appreciate if the authors could expand on it.*

- We have expanded Sec 2.1.3, but the new text does not fully answer the question raised here. For more discussion, see the replies to Adam Sokol and referee #3.

> *Section 4.2: The section is a bit unclear to me and maybe some conclusions are not very easy to reach. The manuscript is already long, but maybe adding some text to the appendix would help readers.*

- We have revised the section, in response to this comment and one by Adam Sokol.

> *I find Figure 9 very interesting. I would be very much interested, if possible, to see what annual cycles other satellite data has (even for shorten time period).*

- It is encouraging that the figure raises interest, as we considered not including it to make the manuscript a bit shorter. However, we decide to not include any other satellite data. As CCIC closely resembles 2C-ICE regarding averages, it can be taken to also represent DARDAR and 2C-ICE. To add an additional dataset for a single figure would cause confusion (and would mean considerable amount of work). In addition, the main message of the figure is maybe not agreement with the observations, but the considerable deviations between the models.

> *General comment on GSRMs: Have you excluded the first 10 days of spin-up following Stevens et al. (2019), Lang et al., (2021), Corko et al. (2025 ...?*

- Yes, that period was excluded. We are just using data from February (that was mentioned). A comment has been added in the introduction of Sec 5 for clarity.

> *In my opinion, it would be preferable if the authors compared the model results with observations at least with respect to the season during which the DYAMOND model simulations were conducted. Despite not being nudged, models were initiated with ECMWF SSTs and therefore the convection is highly influenced with the period of simulations.*

- This is what has been done. The CCIC data are for February 2020, and the satellite uncertainty range in Fig 11 are based on CloudSat retrievals Jan-March 2007-10.

*Section 5.2 (The tropical region): out of curiosity, do you know if convection is more active/stronger during northern summer than northern winter? I would think that this is the case also due to more land over the northern hemisphere. Would that impact the comparison of AC (%) and in particular CC (%) from the winter DYAMOND model and observations from Table 3 (please add to the captions the area and time period where missing)?*

- As already indicated above, differences in time coverage between Secs. 3 and 5.2 need consideration. It is not just a matter of all-year 2015 vs. Feb 2020, but also coverage in local times (1:30 in Sec. 3 and 24h in Sec 5). Here CCIC comes in handy, as it can be used as a common reference. Accordingly, CCIC has been added to Table 7. The results of CCIC in Tables 3 and 7 are not identical, but are sufficiently similar to motivate using results from Sec. 3 for comparison. In fact, the relative poor retrieval performance at high FWP is a more limiting factor (that was noted already in the old manuscript).

*Could you explain the 99.99 percentile? Are those outliers (1 or 2 cases or more)? I am surprised that, in 2C-ICE, it goes to nearly 18kg/m² while with DARDAR to only 9.6 kg/m²? In the convective case you showed in Figure 1, both DARDAR and 2C-ICE go up to nearly 10 kg/m² in the convective core, as well as in figure 2. Can you show the scene of that 99.99 percentile case? Do you know what is so different in DARDAR for extreme cases compared not only to 2C-ICE but also to AOP nominal that constrains DARDAR FWP to cca 10 kg/m² maximum?*

- The statistics in Table 3 are based on about 4.75 million samples. Accordingly, there are about 475 samples above the 99.99th percentile. On purpose, we selected a scene range from thin clouds to FWP values towards the maximum, to illustrate the performance for a broad range of situations.
- We have added a discussion of the retrieval accuracy at high FWP, in Sec. 3.3.

*Line 93: "Comparisons of 2C-ICE and DARDAR are surprisingly few; exceptions include Deng et al. (2013); Winker et al. (2024); Atlas et al. (2024)." It would be great if you cited those papers and their results when you compare DARDAR and 2C-ICE (I feel there is no unique answer regarding the differences between the two products). In my experience, I have always felt they were "less different" than what you show here. Do you have an idea what could be a reason for such a difference? Do you think it would be different if you analysed a different time period? For example, in Atlas et al. (2024), Figure3 shows that in winter period (February data from the years 2007–2012) FWP from DARDAR is substantially larger than in 2C-ICE (opposite to what you found in general (table 4/figure 4 etc...). I feel that generally, this kind of discussions is missing in some parts of the manuscript.*

- We don't discuss in detail why DARDAR and 2C-ICE are different, but we point out main differences in Sec. 2.1.2. They use different ways to separate between liquid and ice, and different particle models. In Sec. 3.5 it is mentioned that both these differences can lead to large systematic differences. Accordingly, we are not surprised that they differ, even in global mean FWP. The question is if there are common biases, causing them both to deviate from the true mean. In Sec 3.6 we argue that this can not be ruled out.

  As discussed above, we have been considering the annual variation of global mean FWP, but we have not looked into the annual variation of FWP distributions. Again an interesting question, and strange that nobody has investigated (to our best knowledge). To include such an analysis in this study would risk distracting the reader from the main points, which are that there are significant differences between satellite retrievals, from local level up to global means. However, we can still point out weaknesses of atmospheric models with the observations. And by that we hope that satellite data will be used even more by the climate modeling community.

  The study of Atlas et al (2024) is restricted to the tropics and in Fig 3 they sub-sample with respect to the presence of high clouds (as we understand it). Accordingly, their Fig. 3 can not be compared directly to our Fig. 3.

**Specific comments:**

> *Line 26-28: "It is noteworthy that the data request document for the latest Coupled Model Intercomparison Project (CMIP6) in effect defines cloud ice as the ice categories considered by the model's radiation scheme." I feel citation would be good.*

- We specified this sentence and now refer to the CMIP6 data request website where the excel sheet with variable definitions can be found.

> *Line 32-33: "Later studies comparing cloud ice from GCMs with various satellite observation include Eliasson et al. (2011); Li et al. (2012); Jiang et al. (2012); Komurcu et al. (2014); Li et al. (2020)." This is an example of only providing citations without any information. Could you provide their results or reason for their mention?*

- As pointed out, the sentence was vague and it has been removed. Most of the references were anyhow used further down in the text.

> *Line 77-78: "In any case, total ice is still the only mass quantity that has a clear definition." Could be expressed better.*

- Now phrased as "In any case, total ice is so far the only mass quantity defined consistently across models and observations."

*Line 78-80: "DARDAR and 2C-ICE have been used as the reference in many studies, but are generally used separately." I think a few citations should be inserted after "many studies".*

- Four examples have been added, two each for DARDAR and 2C-ICE.

*Line 81-82: "DARDAR and 2C-ICE do not offer sufficient coverage for addressing these questions and a dataset representing the state of the art for passive retrievals is also applied (denoted as CCIC)." CCIC has not been defined yet, so please define it. I understand from the text that it is a machine learning (ML) product based on passive observations. Therefore, I would not call it retrieval but how you mentioned ML product.*

- Neither DARDAR and 2C-ICE were defined. However, to properly introduce DARDAR, 2C-ICE and CCIC in this section would cause substantial distraction. Instead, references to the section where they are introduced have been added.
- Please note that we also use "product" together with the other retrievals considered. It is not fully clear if "product" is suggested in favor of "retrieval" because of ML or because passive data has been used. In any case, we argue that that CCIC can be denoted as a retrieval. ML is a new approach, but it solves the same task as traditional methods like OEM (a.k.a. 1D-var). For CCIC (and other ML retrievals) there is a clear formulation of what is optimized, like OEM.

*Line 102-103: "As older passive retrievals have been shown to have a strong bias with respect to CloudSat-based ones, they are excluded from this study." What do you consider as older passive remote sensing? I feel citations are missing.*

- We have changed "older" to "traditional". The sub-sequent paragraph should make clear what we mean with traditional. There were citations, but placed at the start of the paragraph. Now moved into the sentence of concern.

*Section 2.1.3. Passive dataset. Could you explain it a bit more? I am not sure what to expect from this kind of product. IF 2C-ICE has certain amount of ice connected to certain region and season, e.g. over the tropics where passive sensors saturate fast (and will retrieve much lower FWP), machine learning will indicate that there is too low FWP (compared to 2C-ICE) and artificially prescribe statistics from 2C-ICE despite non-physical retrieval relative to passive remote sensing? In that sense, which part is coming from retrieved values from passive remote sensing and which from "copying" the 2C-ICE retrieval? I think it would be helpful to add a few sentences about the product.*

- Section 2.1.3 has been expanded due to comments from all three referees. For your question here, we point out that CCIC is not aware of the location or date. We would rather express it like that the machine learning model has learn to "interpolate" in an very advanced manner between all 2C-ICE retrievals found in the training data.

*Line 190–191: I would replace "resolution" into a grid spacing. "Resolution of 5 km" means that the model grid length is a fraction of that (say, around 1 km or less). Therefore, when one refers to the actual model grid length, then "resolution" should be replaced with "grid length". "Resolution" can be used in a general sense, as in "low resolution models". One can also say "5 km grid" in place of "grid length of 5 km". I think there are a few places, also in the tables, where I feel it needs to be changed to grid spacing.*

- This has been changed.

*Figure 2: The information about the area of analysis is missing.*

- The area used has been added.

*Section 3.3 and text related to Figure 3: To me, it appears that PDFs of FWP (Figure 3a) are very similar and do not vary a lot. In my understanding, everything under 5 g/m² can be detected only by LIDAR, which also loses its sensitivity for <1 or 2 g/m². Therefore, I am not sure if I would focus a lot on the FWP < $10^{-3}$ kg/m². Also, I believe PDF (Figure3a y axis) should not have units. Regarding Figure 3b: could you please explain in more detail how you calculated it? I find values on y axis very low and very different than in the papers you have cited (e.g. Sokol and Hartmann (2020); Atlas et al. (2024).). I believe it is influenced by the number of bins?*

- Correct. A distraction to include bins below 1 g/m2. Figures have been changed accordingly.
- The unit of the PDFs in Fig. 3a should be clear from Eq. 1. Yes, the values in the occurrence fractions in Fig. 3b depend on the bin width, as pointed out in the paragraph below Eq. 2. In the text of Fig. 2 it is mentioned that we have used 200 bins.

*Figure 5: Even though I appreciate the effort to use newly available data, I would still be very careful about using and interpreting EarthCARE data, as it is still experimental and in the validation process. As authors may be familiar with, there will be a few products available regarding the FWP and the product they use is just one of them. Therefore, at this stage, I would rather exclude this experimental data. However, in case the authors want to keep and show it, it has to be clear in the text, caption related to the figure and the legend, that this is still experimental data, yet to*

*be verified, and instead of using "Earthcare", in the legend and elsewhere, the name of the product "CPR_CLD_2A" should be written because it could make an incorrect impression of EarthCARE data.*

- Yes, we should for sure have been more clear about distinguishing between EarthCARE and CPR_CLD_2A. Anyhow, we have now removed CPR_CLD_2A from Fig. 5, following the advice here and after getting a better view of the status of the EarthCARE products through visiting a workshop.

*Line 442-443: "When those outliers were excluded, the remaining models showed differences of about a factor of 6 – comparable to our findings." This result is surprising as it would turn out that there is no overall improvement in the model spread between old CMIP3 and new CMIP6 generations. This is the opposite of what you mention in the abstract. Could you please comment on this and include citations in the results' part that support the progress between the model generations?*

- We removed the part of the abstract that can be interpreted as conflicting with our statement. The sentence in the abstract now reads: "Global circulation models continue to systematically underestimate frozen water paths compared to the observational benchmark and fail to provide a consistent representations of regional temporal changes or the annual cycle."

*Line 448: "Most models participating in the CMIP6 and CMIP6-HighRes underestimate the FWP when compared to satellite retrievals." In my opinion, this is clearly an understatement. Nearly all models in Figure 7 underestimate FWP (except one in each group and some small parts of 3-4 models are barely within the "huge" uncertainty of satellite observations). This might change once it is clear what you mean by satellite uncertainty (see my general comment)*

- We have replaced "Most models" with "Nearly all models" in line with the reviewer's suggestion. We believe that the remainder of the paragraph already provides the information needed to address the main point of the reviewer's comment.

*Line 469-471: "Despite the large variability of FWP in between the models, we can still conclude that none of the models overestimate FWP compared to the satellite retrieval, and therefore none of the models can be falsified. However, it must be considered unlikely that non-reported ice masses can explain the gap to the satellite range for all models." I do not understand what the authors wanted to say here, e.g. "none of the models can be falsified"? Is it always "non-reported" ice as some GCMs include precipitating (snow) ice due to radiation scheme?*

- Here we tried to be very formal, pointing out that there can be non-reported ice masses (even when including "snow" in IWP) that theoretically could move the models up to the satellite range. We pointed out this as very unlikely. Anyhow, the topic is in fact discussed elsewhere. Accordingly, we have simply removed the paragraph, in order to not cause confusion and make the manuscript a bit shorter.

> *Figure 10(11) and related text: Since you are showing CCIC for February in Figure 10, why not include (calculate) its mean value and compare it with that value in DYAMOND models, instead of comparing mean FWP from models with annual CCIC and commenting it is about 3% lower than in February? That way, you would not need to speculate in the 1st paragraph in 5.1 section. In Figure 11, I appreciate the comparison with DARDAR for January to March in the years 2007-2010. Would you also consider adding CCIC zonal mean line from Figure 10? You are working with amazing data and interesting results which could, with minimum additional work, contribute to a very rich analysis (you could also calculate and comment the mean FWP for DARDAR 2007-2010 for January to March which is shown in Figure 11. As expected, it can already be seen in Figure 11 that, even though models mostly underestimate FWP, they are all in the range of DARDAR data when you compare it with the same winter season.*

- A good suggestion to add CCIC's mean to Table 6, simply done and allowed for simplifications in the text.
- For Fig. 11 we prefer to not add more datasets. The figure includes already many lines (bordering on too many). Furthermore, we prefer to not include results from specific retrievals, to not in any way indicate that this dataset stands out (when it comes to giving correct mean values). The full uncertainty range shall be considered.

> *Line 544-546: "Model and observational distributions agree most closely at high occurrence fractions, where they exhibit similar peak values, while most models show higher fractions than the observations around FWP ≈ 10-4 kg m-2." As I already commented before, you are describing FWP of 0.1 g/m², values which observations will most probably not retrieve, not even lidar? Figure 12: Again, I do not understand such small values on y axis. See also my comment 11.*

- This text has been reformulated after removing the range below 1 g/m2. See answer above for the relationship between the distributions values with the selected number of bins.

> *Conclusion (Line 650-654): Here you mention that you showed around 15% spread, in the text for comparison with models, you assume DARDAR data as a reference with 10% spread, but in the abstract "estimate" uncertainty up to 40% (because what Austin et al 2009 found in their paper). And this 40% in the abstract, in my opinion, is misleading of what to expect from the paper since the spread you show is "much" lower. Also, mentioning that EarthCARE data could indicate the 40% uncertainty, based on preliminary, yet to be verified, one product would further mislead the reader and I find it not to be appropriate.*

- We were far from clear here. The value 15% was not actually established in the text, it was just was just a broad comment. This paragraph in the Conclusion builds upon Secs. 3.4 and 3.6, and we now refer to these sections for clarity. And the motivation in Secs. 3.4 and 3.6 regarding the numbers of concern should now hopefully be clear after textual changes in those sections.

**Technical corrections:**

> *A minor point: in several places the manuscript does not clearly distinguish between satellites and the instruments they carry (e.g., CALIOP lidar on CALIPSO, the radar on CloudSat/EarthCARE). This appears to be a writing oversight. Please review the text to ensure that satellites and their respective instruments are correctly identified throughout.*

- Yes, we were not fully clear on this point. Changes have been made to improve clarity.

> *Line 39: GSMRs should be GSRMs*

- Changed.

> *Line 432: "However, we still compare the observations following Waliser et al. (2009), Jiang et al. (2012), Li et al. (2020), and others." – compare models/them with the observations*

- Changed.

---

## Author Comment (AC4)

**Response to review by Anonymous Referee #3**

> *This article reports an intercomparison of atmospheric ice mass, first between different retrievals, leading to an estimated uncertainty within the observations, and then between global climate model simulations as well as global storm-resolving model simulations in reference to these observations. The article is very rich in results. The comparison of atmospheric ice mass is not easy, as the range spans several orders of magnitude, and mean values over such a large range are not enough to fully understand differences. Therefore, distributions of atmospheric ice mass are also shown in observations and in global storm-resolved model simulations. In that way, four datasets, all based (more or less directly) on radar-lidar observations from space, and nine global storm-resolving model simulations are compared.*
> *The comparison between the models is more reported than understood, in particular for the global climate models, but this reporting is a first step so that the different model teams can work on further improvement of the parameterizations.*
> *As the outcome of this assessment is important, I recommend publishing, but only after a major revision. This revision is mostly to improve the structure of the article and to clarify certain points, so it should not be too difficult for the authors.*

- Many thanks for finding our assessment important, pointing out some of the challenges, and providing helpful comments. We have done our best to clarify uncertain points. However, we have not changed the structure of the manuscript, a decision motivated below.

- Please, find below replies to your comments. We gave emphasis to points where there are commonalities with the input from one or both of the other referees, still carefully considering all points raised.

**Major comments**

> *The introduction explains well the problems in the definition of cloud ice. However, it is not completely clear if the final term 'frozen water path (FWP) corresponds to a grid-box average or an in-cloud average. From Pfreundschuh et al (2025) I deduce that it is the grid average. This should be clearly stated. The in-cloud FWP, which is directly retrieved by DARDAR and 2C-ICE, is also interesting to compare, as it is used in the cloud radiative transfer in the models.*

- Many thanks for this important remark. We missed this aspect despite ourselves occasionally struggling, when reading journal articles, to understand if "in-cloud" or "all-weather" averages are displayed. We use all-weather (grid-box) averages. This is now clarified, just after defining FWP (towards the end of the Introduction), and is repeated in the Conclusions.

*The overarching goal of this article seems to be the assessment of the simulated atmospheric ice mass, but the authors also took the effort to intercompare four different datasets, based on satellite retrievals using radar–lidar or radar–only observations. In particular, they present results of the relatively new CCIC dataset, which is based on Machine Learning (ML) techniques trained on the CloudSat–lidar 2C–ICE product. So, the goal is actually two–fold. This should be more clearly formulated in the abstract and in the introduction.*

- We really had the two–fold goal in mind, to both assess observations and models. To make this clearer, observations is now placed before models in title, the start of the abstract is rewritten, and additions have been made at the end of the Introduction.

*Though several publications exist about this new dataset, it would be very helpful to clarify the description of this dataset and to show the uncertainty of this dataset which comes out of the applied Machine Learning technique, as explained in one of the earlier publications. More detailed questions and comments on this issue:*
*(a) In Pfreundschuh et al. (2025), the used ML technique for the CCIC is a convolutional neural network (CNN), while in this article the retrieval is given as quantile regression neural networks (QRNN). This is confusing. Indeed, the authors cite several articles which describe the retrieval, but it would help to give a more detailed overview of this retrieval. I am very surprised how with only the use of one 11 micron brightness temperature (TB) together with the structure of the TB variability over regions of about 900 x 900 km2 (with 256 x 256 pixels) allows for such accurate prediction of IWP of the CloudSat–lidar 2C–ICE product, the latter given on a spatial resolution of about 1.5 x 2.5 km2. The TB depends on cloud height, on ice crystal habit and size distribution and on IWP. The TB also depends on season and daytime. How is this taken into account, in particular when the data are also expanded to other observational times than 1:30 AM and 1:30 PM LT?*

- Sec 2.1.3 has been expanded to provide more information on CCIC. For example, it shall now hopefully be clear that CCIC combines CNN and QRNN. However, to not introduce a poor balance with respect to the other retrievals, we still try to keep the description of CCIC compact. That is, we don't want the CCIC to overshadow DARDAR and 2C-ICE.
  We can not really explain why machine learning is managing so well in these retrievals, except concluding that there must be patterns that we humans still are unaware of, but the neural network has managed to identify. On the other hand, in the new text we indicate why CCIC also works outside of the CloudSat local times. In short, the retrievals should work acceptably at e.g. 6:00 PM if something resembling the local cloud situation is found in the training data, obtained at 1:30AM or 1:30 PM. Please remember that we do not make use of any optical or NIR channels and thus are not sensitive to the solar angle.

*(b) In general, regression neural networks give the right average compared to the dataset they are trained on, but scene-dependent biases exist when scenes with very large IWP are rare, as is the case in the tropics. This effect is even larger when the retrieved variable spreads out over several orders of magnitude. This reduces then the range of the ML-derived variable, as can be seen in Fig. 6 of Amell et al. (2025) or in Fig. 5 of Pfreundschuh et al. 2025. Somehow these biases show in the difference between the distributions in Figures 3. There is a large part with very small FWP, can the authors explain these cases?*

- We agree with these observations. CCIC tends to over/under-estimate when the true FWP is below/above about 1 kg/m2, as shown in earlier works and here in Fig 2. And these tendencies give CCIC the highest values in Fig 3 at lower FWP. As discussed just below, still CCIC manages to represent the retrieval uncertainty through providing data to establish confidence intervals.

*(c) Indeed, the results are much better than those using only passive remote sensing, but it would be interesting to see the uncertainty of the ML retrieval. In Pfreundschuh et al. (2018), it is written that QRNNs also provide the uncertainty, but I do not see this uncertainty quantified or presented in the current manuscript.*

- CCIC uncertainty estimate is now exemplified in Fig. 1. Please note the start of new text in Sec. 3.6, explaining that the uncertainty estimates are difficult to apply for averages, independently of whether CCIC, DARDAR, or 2C-ICE is used.

*The structure of the article:*
*(a) After section 2 (Data) which presents Satellite retrievals and models, it is confusing to see sections 3 Satellite retrievals, 4 GCMs and 5 GSRMs. I would include section 3 'Intercomparisons' and then put the initial sections 3–5 as subsections: 3.1 Satellite retrievals, 3.2 GCMs and 3.3 GSRMs.*

- We have not changed the structure as we see this as two-stage process, that we first establish the retrieval trueness in Sec. 3, and then move on to assess the models. The start of the abstract has been rewritten and a paragraph has been added at the end of the Introduction, to be clearer about that the second part builds upon the first. To this we add that none of the other referees have suggested changes in this direction.

*(b) Furthermore, it is very confusing to see an outlook (section 3.7) in the middle of an article. Normally the outlook comes after the conclusion of a scientific article, which itself presents scientific results and their interpretation.*

- Using "Outlook" was a poor choice of name for the section. Now called "Emerging satellite retrievals".

*(c) The interpretation of Figure 5 needs some clarifications: The CCIC results are now shown for 10:30 and 22:30 LT, while they have been obtained via ML with a training at 1:30 and 13:30 LT. There is not one sentence on the reliability of this expansion in time. Also, what exactly is the satellite uncertainty shown in gray in Fig. 5? Another interesting point is that CCIC and SPARE-ICE show very similar zonal averages (except NH subtropics). Does this mean that the microwave information is useless in the retrieval of IWP (as CCIC only uses one IR channel)? Is it possible to give some explanations? Also, the authors state that the EarthCARE sensor and retrieval are improved. As the EarthCARE zonal mean is quite low in the tropics, does this mean that the high peaks in CCIC and SPARE-ICA and AWS are due to not-detection of thinner cirrus? This seems to be a huge effect.*

- The low mean of EarthCARE CPR_CLD_2A in the tropics was explained in the text, but could easily be missed. Anyhow, no EarthCARE product is now included, following a suggestion by Karol Ćorko. The figure is thus updated, and now including CCIC for both July and August.

- The validity of CCIC outside of 1:30 and 13:30 LT is commented on above. We also want to draw the attention to that we cited Leko (2025), a master thesis report showing that CCIC indeed provides realistic diurnal variations of FWP, that compare well with the DYAMOND models (or the reversed?). Based on these promising results, we have now started a manuscript on the subject, including comparison to the diurnal variation of precipitation (as given by IMERG), that we hope to submit to a journal relatively soon.

- Regarding the role of microwave data in SPARE-ICE, we remind that a retrieval can have poor precision but still have a good trueness (like CCIC). That is, a less precise retrieval can still provide good averages. We have not looked at this specifically for SPARE-ICE, but we see this clearly when comparing local values from AWS and CCIC. The AWS retrievals contain more horizontal structures and has a higher "dynamical range". So yes, we strongly believe that also passive microwave observations can contribute to FWP measurements.

*(d) Many intercomparison results are shown, but for example to compare the global mean of a variable which spans several orders of magnitude is not a strong assessment. One interesting point here is that the IFS distribution (Fig. 11) does not agree with the observations, but the near-global mean does! Since the intercomparison sections are quite long, one could probably take the comparison of the global means to the supplementary material and include the global mean values to Table 1 which could also be moved to a supplement, and then one starts this section with the comparison of zonal means. The same for the global means of the GSRMs: I suggest combining Table 6 with Table 2 and moving them to the supplementary material.*

- As argued above, we see this is a two-stage process, starting with the observations and moving on to the models. Unfortunately, we failed in the original manuscript to make that clear. Anyhow, a rearrangement as suggested would indicate that we would look on observations and model results equally. We strongly argue for that the observations shall be seen as a reference for the models.

- We think that global means provide a broad overview of the models performance on FWP, and thus constitute a good start in both Secs. 5 and 6. But yes, global means do not give the complete picture, as IFS exemplifies. However, good correct zonal means can be achieved by an incorrect distribution of FWP. If the FWP distribution behind a zonal mean is fair, it can still be bad for certain longitudes ... We start at the broadest level and zoom in one or two steps, that is all that can accommodated in a single journal article. We encourage others to take the assessment further.

> *4. Retrieval trueness and estimated uncertainty in section 3.6:*
> *(a) I have difficulties to follow the argumentation. From Fig. 1 it looks like 2C-ICE seems to be more sensitive to thin Cirrus and therefore the distribution in Fig. 3 shows two peaks. 2C-ICE also seems to have a larger range in FWP towards larger FWP. Since the range towards the larger FWP counts more in the mean than the larger range towards smaller FWP, the authors find a 24% larger mean. Why should you put more weight on DARDAR and AOP, the latter only using CloudSat data?*

- Yes, Sec. 3.6 was far from clear. Section 3.6 is totally rewritten and some critical additions have been made to Sec. 3.4.

> *(b) The uncertainty range of 40% is assumed without any further explanation, and this is highlighted as result in the abstract. Why do you not show the uncertainty of CCIC which you claim in earlier articles can be obtained via QRNN? The sensitivity studies in section 3.5 show another part of uncertainty, based on the microphysical assumptions. You could base your argumentation on these findings.*

- We did not involve CCIC in Sec. 3.6, the errors stated in the DARDAR and 2C-ICE products would have been more relevant. Anyhow, we now start the section by pointing out that local uncertainty estimates (as provided today) unfortunately are hard to map to errors for averages. Section 3.5 was added with the "retrieval trueness" discussion in mind. We ended up to using it more as background information. The new version of Sec. 3.6 refers more directly to the results in Sec. 3.5.

**Minor comments**

> *Title: 'ice mass' instead of 'ice masses'?, same in line 11*

- A good suggestion. We did not consider that mass here can be pluralized as mass. Title changed. We have also changed in the text, but kept masses in some places where it can help the understanding.

> *p 1, l 6 –7: 'but its accuracy is limited by biases inherited from its training dataset' : it is true that ML can as best be the same as the training dataset and therefore naturally includes its biases. However, this is trivial, and I would like to see in the abstract also mentioned the additional biases and uncertainty linked to the reduced input.*

- Correct, that phrasing was not very informative. The sentence now reads: "A recently developed machine learning product based on passive thermal infrared observations highly extends spatial and temporal coverage for comparisons, but its local precision is limited compared to radar-based retrievals."

> *p 4, l 93: you may add Vidot et al. 2015 (DOI: 10.1002/2015JD023462), they compared IWC profiles for small and large COD (Fig. 4).*

- We have changed the text (e.g. adding "in depth"), and as DARDAR and 2C-ICE are used together in many studies the word "surprising" has been removed. To reflect this we have also added some references. To do this in "objective" manner, we asked Google Scholar Labs to list the three studies that have performed the most in-depth comparison of DARDAR and 2C-ICE, and we added the two we had not already cited.

> *Section 2.1.2: I would move the second paragraph (p 5, l 120–123) to the front of this section*

- Yes, a better order. The suggestion has been implemented.

> *p 4, l 116: take out 'retrieval'*

- Done.

> *p 5, l 143: 'radar bin' perhaps 'radar vertical segment' ? is each bin or vertical segment about 0.5 km ?*

- Yes, "bin" not the best word. We have changed to use "range gate", or just "gate". Yes, the size of the ranges are 500 m.

> *p 7, l 6 & 7: please add 'boreal' in front of 'Summer' and 'Winter'*

- Boreal has been added.

> *p 7, l 210-211: we sum up ... (IWP, GWP, SWP): is this weighted by their fraction within the grid ?*

- With the addition to the Introduction, that we solely use all-weather means, this should now be clear. Adding a remark here could rather have the opposite effects. It could be taken as that in-cloud averages are used in some place.

> *p 19, l 434-435 Section 4: 'the overall assessment is based on global means': This is really a pity, but probably CMIP6 results only provide the monthly means? It would be important to add in the conclusions that distributions should be added as output for CMIP7. However, you need also to mention that the distributions in Figures 3 may change their shape when reducing the spatial resolution to 100 or 250 km. Did you have a look how they would change?*

- CMIP6 provides also daily and 3-hourly clivi outputs, but these are available for a smaller subset of models compared with the monthly mean outputs. Therefore we consider the monthly mean outputs to be more suitable for our work, since we wanted to include as many models as possible. Furthermore, the usage of 3-hourly, daily or monthly outputs should not produce a remarkable change in the results. Regarding Figure 3, it does not include CMIP6 models. For this reason it is unclear to us why it is referenced in this context.

> *Comparison of zonal means (Fig. 7): the authors compare grid averages of FWP from the model simulations to the range in satellite observations coming from nadir tracks; how do the authors build grid averages if there is only a narrow track within a grid of the GCM spatial resolution? Here, actually the CCIC dataset may be useful as it is expanded to fill a whole grid, even though additional uncertainty is added due to ML expansion.*

- The narrow across-track coverage of the CloudSat results in relativly "noisy" local averages (for e.g. 1x1 lat/lon grids), but we are not showing such averages. For annual zonal means each latitude bin contains enough of samples to provide relatively stable statistics (otherwise the zonal means would be less smooth). The CloudSat measurements are equally distributed in longitude, and averaged they will give a fair approximation of the true zonal means.

> *Section 5: Why do you limit the GSRM means to 60N-60S while the GCMs are averaged over 90N-90S?*

- We limit the analysis of the GSRMs to 60N-60S because our goal is to compare them with CCIC, which is limited to these latitudes. For consistency we have now updated the GCMs analysis (including figures) to also cover only the 60N-60S latitude range.

*Figure 10: instead of (or in addition to) comparing the mean FWP of CCIC and the 9 models, one could show the difference map between both estimations in order to see where there may be differences.*

- The aim of the figure is to introduce the general capacity of the GSMR models, that it is high in comparison to the GCMs. Representing the GSRMs by their difference to CCIC would instead put emphasis on the differences. In any case, the differences to CCIC are hard to analyze for the ensemble mean, they are more relevant for individual models.

*Figure 16: It is known that the diurnal cycle of convection differs over ocean and over land, therefore a comparison seems only to make sense when ocean and land are separated.*

- Thanks for this suggestion, that we have followed. The figure and text have been updated accordingly.

*Section 5.3:*
*It is interesting that the authors also explore convective indices, but it is difficult to follow this section.*
*p 31, l 604–605: 'In particular, they conclude that the use of multiple indices is advantageous to successfully characterize the underlying organizational structure.'*
*For me, it looks like they concluded first that several of these indices don't fulfil certain quality criteria, like sensitivity to noise under certain conditions, to spatial resolution etc. and this can explain differences in conclusions about convective organization when using different indices; and second that these indices may not be enough to completely characterize organization. Another conclusion was that some indices are highly correlated with one simple variable, like ABCOP reflects the total area of convective objects, while ROME is very strongly correlated with the mean size of the objects. The latter can be seen by comparing Fig. 15 with Table 8. SCAI and MSCAI strongly depend on the number of objects, which may be very noisy. Since Iorg does not consider the size of the objects, the conclusion on organization using Iorg and ROME does not agree. Perhaps one can add in the table the correlations with the corresponding variable, and it would also be good to highlight in bold or italic the largest and smallest for each index.*

- We have added highlighting in the table, as suggested. The cited sentence seems to have given a wrong impression of the sub-section's aim. It is now removed, but other text has been added to be clearer about the aim, as well as limitations. Our purpose was not to identify a best index. Rather the opposite, despite testing multiple indices, we find none that gives an agreement between the models. We think this is an interesting finding, considering the present strong interest in convective organization.

> *Conclusions:*
> *According to the questions and comments before, some parts need to be rewritten, in particular the2. paragraph p 32, l 652 – p33, l 654.*
> *p 33, l 655: 'Global climate models remain biased low': it may be good to add here that this is on grid average; it can be different in in-cloud IWP (see above).*

- We now remind about the all-weather averaging at the beginning of the Conclusions.

> *p 35, 653-654: I do not understand the last sentence of the paragraph. Why indicate preliminary results from new sensors, for which the retrieval is also based on assumptions indicate the pessimistic view?*

- Yes, that argumentation was not clear. The sentence has been removed.

> *p 35, l 721-722: I do not see any Figures B2 and B3 in the newest manuscript.*

- Please note the hint in our comment (AC1), that some action can be needed to make the browser to download the new version.

**Typos**

All "typos" have been fixed. Thanks for spotting these mistakes.